# Large vesicle extrusions from *C. elegans* neurons are consumed and stimulated by glial-like phagocytosis activity of the neighboring cell

Yu Wang[1], Meghan Lee Arnold[1], Anna Joelle Smart[1], Guoqiang Wang[1], Rebecca J Androwski[1], Andres Morera[1], Ken CQ Nguyen[2], Peter J Schweinsberg[1], Ge Bai[1], Jason Cooper[1], David H Hall[2], Monica Driscoll[1], Barth D Grant[1,3]*

[1]Department of Molecular Biology and Biochemistry, Rutgers University, Piscataway, United States; [2]Department of Neuroscience, Albert Einstein College of Medicine, Rose F. Kennedy Center, Bronx, New York, United States; [3]Rutgers Center for Lipid Research, New Brunswick, United States

*For correspondence:
barthgra@dls.rutgers.edu

Competing interest: The authors declare that no competing interests exist.

**Abstract** *Caenorhabditis elegans* neurons under stress can produce giant vesicles, several microns in diameter, called exophers. Current models suggest that exophers are neuroprotective, providing a mechanism for stressed neurons to eject toxic protein aggregates and organelles. However, little is known of the fate of the exopher once it leaves the neuron. We found that exophers produced by mechanosensory neurons in *C. elegans* are engulfed by surrounding hypodermal skin cells and are then broken up into numerous smaller vesicles that acquire hypodermal phagosome maturation markers, with vesicular contents gradually degraded by hypodermal lysosomes. Consistent with the hypodermis acting as an exopher phagocyte, we found that exopher removal requires hypodermal actin and Arp2/3, and the hypodermal plasma membrane adjacent to newly formed exophers accumulates dynamic F-actin during budding. Efficient fission of engulfed exopher-phagosomes to produce smaller vesicles and degrade their contents requires phagosome maturation factors SAND-1/Mon1, GTPase RAB-35, the CNT-1 ARF-GAP, and microtubule motor-associated GTPase ARL-8, suggesting a close coupling of phagosome fission and phagosome maturation. Lysosome activity was required to degrade exopher contents in the hypodermis but not for exopher-phagosome resolution into smaller vesicles. Importantly, we found that GTPase ARF-6 and effector SEC-10/exocyst activity in the hypodermis, along with the CED-1 phagocytic receptor, is required for efficient production of exophers by the neuron. Our results indicate that the neuron requires specific interaction with the phagocyte for an efficient exopher response, a mechanistic feature potentially conserved with mammalian exophergenesis, and similar to neuronal pruning by phagocytic glia that influences neurodegenerative disease.

## Editor's evaluation

This article will be of interest to a wide range of cell biologists working at understanding cell–cell communication. The authors present compelling data showing that large extrusions (exophers) of neuronal cells are taken up by adjacent hypodermal cells and eventually degraded by lysosomes, uncovering an important mechanism for clearing toxic cargo. Mechanistically, the study identifies a number of small GTPases and accessory components, as well as the phagocytic receptor (CED-1).

## Introduction

Cells expend a great deal of energy and resources to maintain the quality of their active proteome, matching expression with degradation, sensing and regulating protein folding and protein complex assembly via chaperone systems, and employing several types of protein degradation systems to remove damaged or aggregated protein products (*Labbadia and Morimoto, 2015*). As long-lived cells that typically cannot divide, neurons may be especially vulnerable to loss of proteostasis equilibrium, with protein aggregation proposed as a common element in the pathophysiology of several prevalent neurodegenerative diseases (*Kurtishi et al., 2019*). In some cases, transfer of neurotoxic protein aggregates and damaged organelles to neighboring cells has also been demonstrated, potentially acting to ameliorate the deleterious effects in the originating neuron, but in turn such transfer may be deleterious to receiving cells (*Davis et al., 2018*). How such transfer processes contribute to the etiology and pathology of neurodegenerative diseases such as Alzheimer's, Parkinson's, and Huntington's is an area of increasing investigation (*Davis et al., 2018*).

Recent studies indicate that wholesale ejection of aggregated protein and damaged organelles into extracellular vesicles represents another pathway to deal with toxic proteostress (*Melentijevic et al., 2017*; *Nicolás-Ávila et al., 2022*). Of particular interest are giant extracellular vesicles called exophers, with diameters rivaling that of the neuronal somata from which they bud, first discovered in *Caenorhabditis elegans* neurons under proteostress (*Melentijevic et al., 2017*; *Nicolás-Ávila et al., 2020*). Ejected exophers can carry aggregates and mitochondria out of the neuron, and the production of exophers is greatly stimulated under stress conditions that promote accumulation of misfolded protein (*Melentijevic et al., 2017*; *Cooper et al., 2021*). For example, exopher production is greatly increased by inhibition of proteosome and autophagy activity, increased osmotic strength, increased oxidative activity, nutrient deprivation, and forced expression of aggregating Hungtingtin polyQ or mCherry proteins (*Melentijevic et al., 2017*; *Cooper et al., 2021*). In the case of Huntingtin HTT-Q128 protein expression, neurons that produced exophers maintained better neuronal function than those that did not, suggesting that exopher production is protective to neurons experiencing proteotoxic load (*Melentijevic et al., 2017*). Importantly, recent work indicates similar mechanisms operate in mammalian systems, with exophers reported to export oxidized mitochondria from highly active mouse cardiomyocytes (*Nicolás-Ávila et al., 2020*), and exopher-resembling giant vesicles identified in human and mouse brain, with an apparent increase in exopher number in diseased brain samples (*Siddique et al., 2021*).

The best studied system for analysis of neuronal exopher production remains the six *C. elegans* mechanosensory touch receptor neurons where exophers were discovered (*Melentijevic et al., 2017*; *Arnold et al., 2020*). These neurons are largely unipolar, each extending a long sensory neurite embedded in the hypodermis (skin) of the animal where each senses gentle touch to the animal body region within its receptive field, leading to a rapid retreat behavior when stimulated (*Bounoutas and Chalfie, 2007*). Of the six touch neurons in the hermaphrodite, the centrally located ALMR neuron produces exophers with the highest frequency and is the main model used in our studies (*Melentijevic et al., 2017*; *Cooper et al., 2021*; *Arnold et al., 2020*). Previous work showed that 10–20% of ALMR neurons stimulated by high-level expression of mCherry produce an exopher (*Melentijevic et al., 2017*; *Arnold et al., 2020*). If an ALMR neuron produces an exopher, it generally only produces a single such vesicle in its lifetime, with most exophers produced on day 2 of adulthood (*Melentijevic et al., 2017*; *Arnold et al., 2020*). Very few exophers appear outside of the day 1 to day 3 timeframe, which correlates with the high metabolic activity associated with reproduction, and an organismal switch in overall proteostatic stress response (*Labbadia and Morimoto, 2014*; *Morimoto, 2020*). Exophers in the ALMR neuron typically emerge from the neuronal soma in a polarized manner, formed from the plasma membrane opposite the large neurite (*Melentijevic et al., 2017*; *Arnold et al., 2020*). Interestingly, as a budded exopher moves away from the soma it initially remains connected to the soma by a thin thread-like nanotube (*Melentijevic et al., 2017*; *Arnold et al., 2020*). Ultimately the connection to the soma is lost and the exopher is released. The exopher quickly breaks up into smaller vesicles that we refer to as 'starry night' (SN) vesicles, which later disappear (*Melentijevic et al., 2017*; *Arnold et al., 2020*; *Figure 1A*).

It is of critical importance to understand the fate of the exopher and the material it carries once released from the neuron. The *C. elegans* system offers the opportunity to understand this process in the native cellular context of the intact animal in which neurons are constantly interacting with

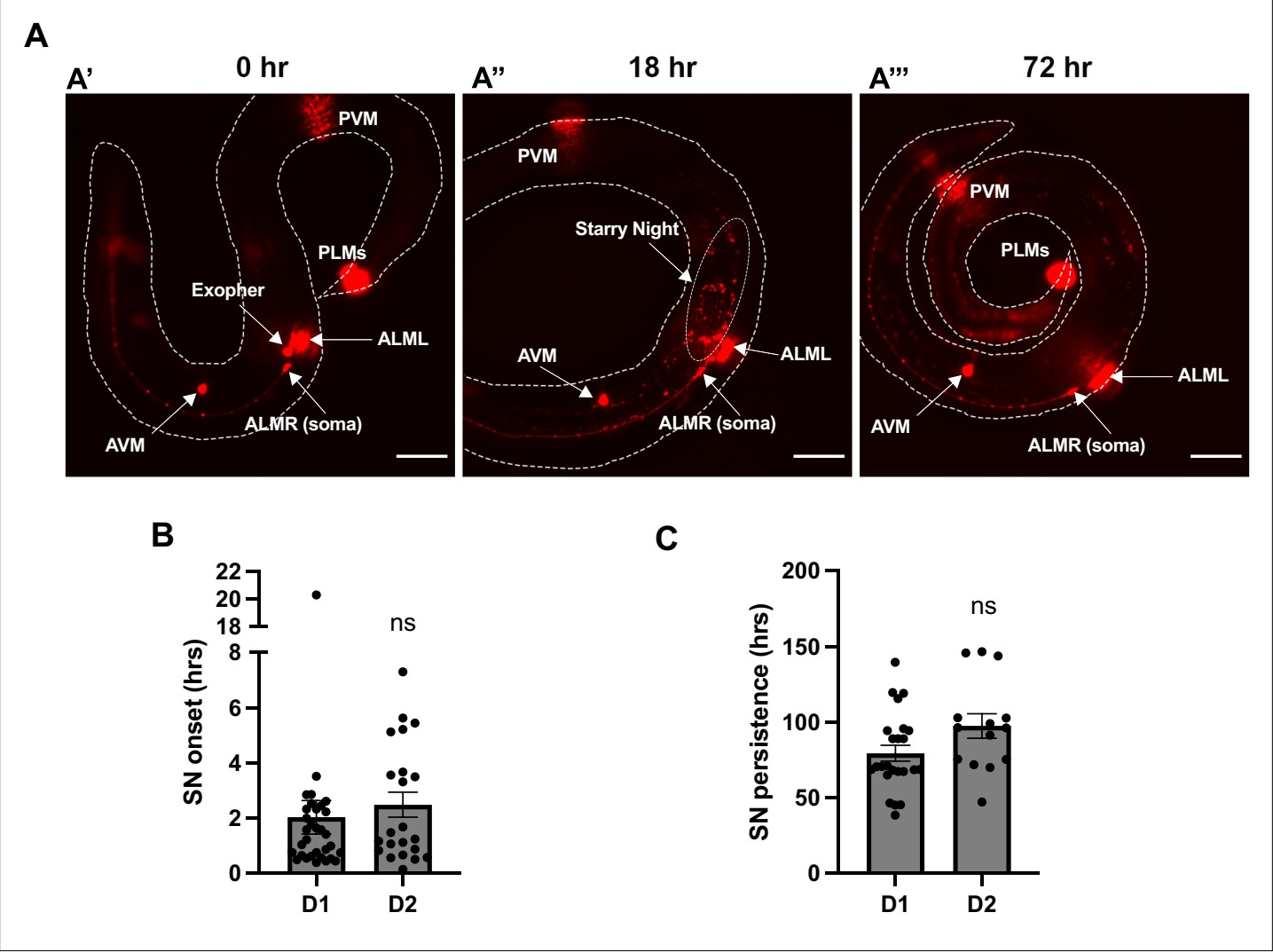

**Figure 1.** Longitudinal analysis indicates that exophers vesiculate prior to degradation of exopher-derived cargo. (**A**) Sequential images captured on a fluorescence dissecting microscope are shown for the same animal at different timepoints. The six touch receptor neurons are marked by mCherry expression. Images focus on the ALMR neuron near the center of the body. The images show exopher production (**A'**), followed by exopher vesiculation (**A''**), and finally loss of exopher-derived mCherry signal (**A'''**). ALMR, anterior lateral microtube neuron right; ALML, anterior lateral microtube neuron left; AVM, anterior ventral microtube neuron; PVM, posterior ventral microtube neuron; PLM, posterior lateral microtube neuron. (**B**) Time from exopher identification to first observation of exopher vesiculation is graphed. Each data point represents an individual tracked animal. Data from adult day 1 (D1) exophers and adult day 2 (D2) exophers are graphed separately. SN = starry night. (**C**) Time from the start of exopher vesiculation to the observed loss of exopher-derived mCherry signal is graphed. Each data point represents an individual tracked animal. Data from adult day 1 (D1) exophers and adult day 2 (D2) exophers are graphed separately. Scale bar = 50 µm.

The online version of this article includes the following source data and figure supplement(s) for figure 1:

**Source data 1.** Numerical data for *Figure 1*.

**Figure supplement 1.** Three exopher zones.

neighboring cells. Here we show a key role for the closely associated hypodermis in phagocytosis and degradation of neuronal exopher contents. The acquisition of hypodermal phagosome maturation markers by the exopher-laden phagosome was low until phagosome fission into smaller starry night vesicles occurred, with lysosome activity required for the degradation of starry night vesicle content after phagosomal fission. Importantly, our data also indicate a role for hypodermis-neuron interaction beyond simple exopher degradation, as the ability of the neuron to produce exophers appears to depend upon action of certain regulators in the hypodermis, including the GTPase ARF-6, its effector SEC-10/exocyst, and the phagocytic receptor CED-1/Draper.

## Results

### Exophers break up into smaller vesicles after release from the neuron

As previously described, exophers appear most frequently on adult day 2 (D2), with fewer appearing on adult day 1 (D1) or adult day 3 (D3), and very few appearing outside of this age range (*Melentijevic et al., 2017*; *Arnold et al., 2020*). To better understand the fate of exophers after they are produced, we first performed manual longitudinal tracking in individual animals focused on D1 or D2 exophers produced by the ALMR touch receptor neuron expressing a touch neuron-specific mCherry marker (*Figure 1A*). For these experiments, we followed animals in their unperturbed and unrestrained culture plate environment using a high-magnification epifluorescence-dissecting microscope.

We noted that the vast majority of mCherry-filled exophers convert into smaller 'starry night' (SN) vesicles, and this always precedes the eventual loss of the neuronal exopher-derived mCherry signal (*Figure 1A–C*). Transient tubulation and rapid vesicle movement is a prominent feature of exopher vesiculation (*Video 1*). Exopher-derived small vesicles often appeared within the first 2 hr after exopher formation (D1 mean = 2.0 hr [N = 32], D2 mean = 2.5 hr [N = 22]). Most exophers were eventually fully converted to starry night vesicles, with all detectable exopher-derived mCherry signal lost over the course of 3–4 days (D1 mean persistence = 80 hr [N = 24], D2 mean persistence = 98 hr [N = 14]). A minor fraction of exophers did not convert to starry night vesicles (7%, N = 118). These exophers persisted and were never degraded over a 7-day tracking period, apparently failing to engage the hypodermal degradative machinery. Taken together our results suggest an obligatory transition from large exopher to smaller starry night vesicles prior to eventual degradation of exopher contents in the hypodermis.

### Neuronal exophers undergo phagocytosis by the hypodermis

Large particles such as apoptotic bodies are engulfed and degraded by phagocytosis (*Levin et al., 2016*; *Ghose and Wehman, 2021*). Once a forming phagosome seals, the process of phagosome maturation begins, eventually leading to the degradation of the phagocytosed material. Phagocytic cups are characterized by polymerization of abundant F-actin that supports plasma membrane deformation to surround the target of engulfment. Phagosome maturation proceeds via sequential fusion of a sealed phagosome with endocytic compartments, generally beginning with early endosomes, then late endosomes, and finally lysosomes. Recycling of material not destined for degradation, such as phagocytic receptors, proceeds at the same time. To determine whether mCherry-containing exophers undergo phagocytosis and phagosome maturation in the surrounding hypodermis, we expressed a variety of molecular markers with mNeonGreen tags from single-copy transgenes using a hypodermis-specific promoter (Phyp7) from the *semo-1/Y37A1B.5* gene (*Philipp et al., 2022*).

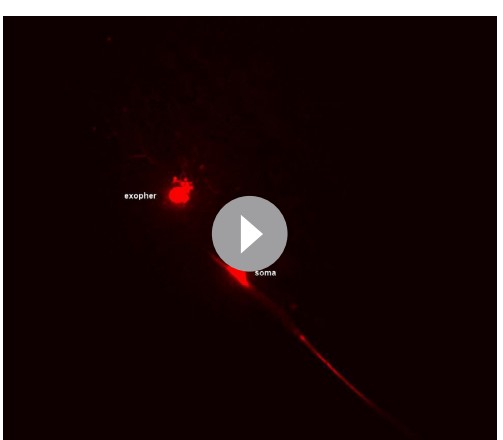

**Video 1.** Exophers tubulate and vesiculate to form starry night vesicles. mCherry-filled ALMR neuron-derived exopher tubulates and vesiculates after release from the neuron. Video depicts a 3 hr 17 min time course, 5 min per frame. Scale Bar = 3 µm.
https://elifesciences.org/articles/82227/figures#video1

We focused first on intact early-stage exophers that had not started to break up into the smaller starry night vesicles. If the exopher is engulfed by the surrounding hypodermal cells via phagocytosis, we would expect hypodermal F-actin to surround the extruded exopher during the initial stage of exopher processing. To test this model, we expressed F-actin biosensor mNG::UtrCH, encoding an mNeonGreen fusion to the CH-domain of Utrophin (*Winder et al., 1995*), specifically in the hypodermis. We then quantified the F-actin signal at the periphery of newly formed exophers marked by mCherry (*Figure 2A and B*). As a control, we performed the same analysis on the ALMR neuronal soma from which the exopher was derived since the soma is also surrounded by the hypodermal Hyp7 cell (*Bounoutas and Chalfie, 2007*). We found that most exophers we examined were clearly coated in hypodermal F-actin (74% positive, N = 27), while the hypodermal membrane around the neuronal soma was

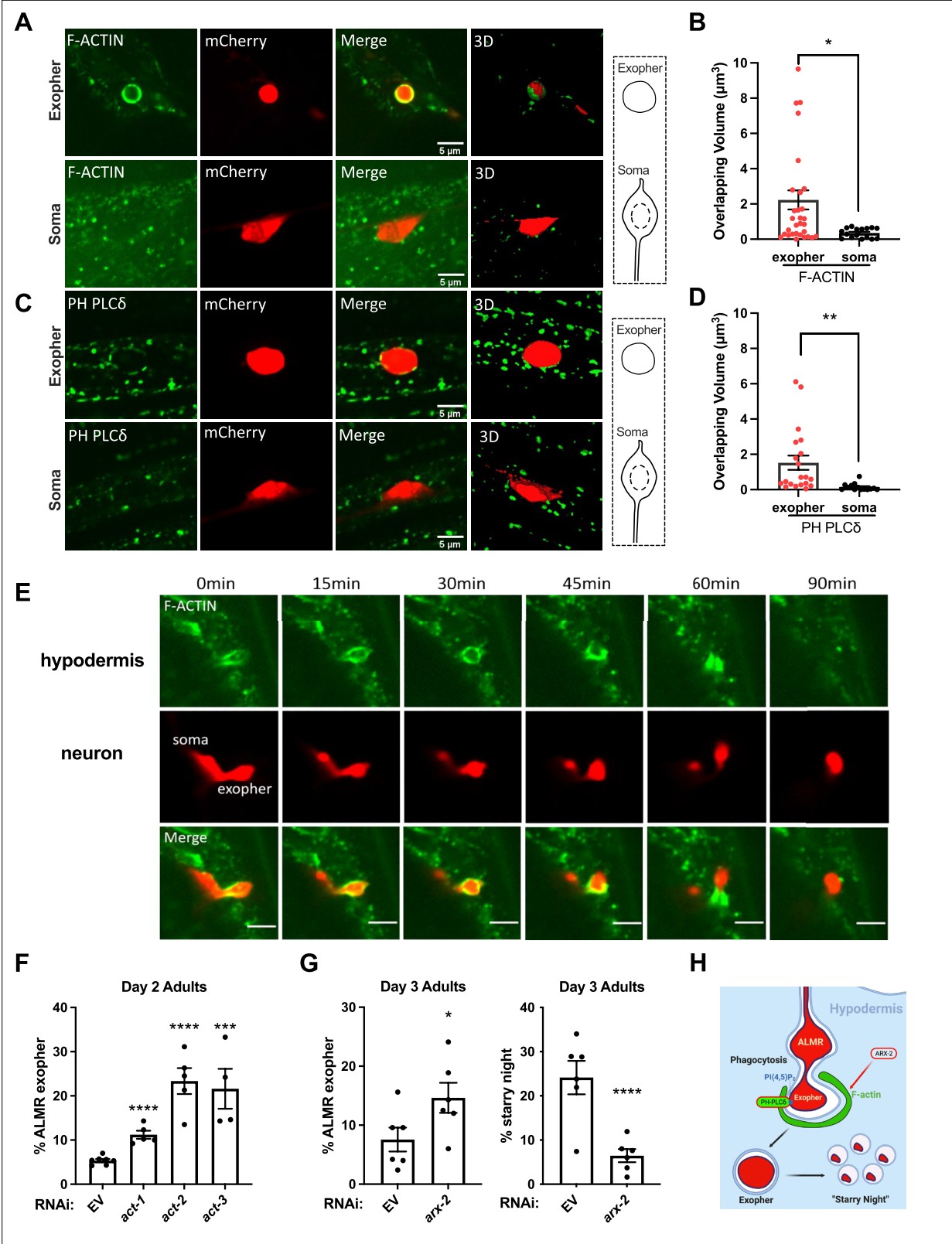

**Figure 2.** Neuronal exophers are phagocytosed by the adjacent hypodermis in an F-actin-dependent manner. (**A, C**) Confocal fluorescence micrographs are shown for an mCherry-labeled ALMR neuron-derived exopher or the similar-sized mCherry-labeled ALMR neuronal soma from the same neuron. Surrounding hypodermis-specific expression of UtrCH::mNeonGreen (an F-actin biosensor) and PH(PLCδ)::mNeonGreen (a biosensor for the lipid PI(4,5)P2) expressed from the hypodermis-specific *semo-1* promoter are shown. A merged image and a merged 3-D projection are shown for each

*Figure 2 continued on next page*

*Figure 2 continued*

example. Scale bar, 5 µm. (**B, D**) As a measure of hypodermal marker recruitment to the exopher, we measured the volume of overlapping signal in 3-D projections between the hypodermal marker and ALMR-neuron-derived exopher, comparing to the ALMR neuronal soma as a control, *p<0.05 by two-tailed unpaired *t*-test. N = 24 (F-actin) and N = 18 (PI(4,5)P2). (**E**) Time-lapse images of hypodermal F-actin dynamics during the engulfment of the exopher. '0 min' indicates the beginning of observed time course. Scale bar, 5 µm. (**F**) Histogram depicting percentage of exopher-positive animals upon hypodermal-specific RNAi for empty vector control (L4440), *act-1*, *act-2*, or *act-3* in day 2 adults. ****p<0.0001, *p<0.05 calculated using the Cochran–Mantel Haenszel (CMH) test. Each point represents a trial of n = 50 animals scored, with four trials per condition. (**G**) Histogram depicting exopher and starry night-positive animal frequency after hypodermal-specific RNAi for empty vector (EV) control, or *arx-2* (ARP2/3 subunit) in day 3 adults. ****p<0.0001, *p<0.05 calculated using the CMH test. Each point represents a trial of N = 50 animals scored, with six trials per condition. (**H**) Diagram summarizing interpretation of events, including branched actin-dependent engulfment of ALMR-neuron exophers by the hypodermis, a prerequisite for later vesiculation of the completed phagosome.

The online version of this article includes the following source data for figure 2:

**Source data 1.** Numerical data for *Figure 2*.

not enriched in F-actin (*Figure 2A and B*). We also noted that most (65%, n = 45) intact exophers were surrounded by hypodermal puncta positive for mNG::PH(PLCδ), a biosensor for the lipid PI(4,5)P2, with significantly more PI(4,5)P2 signal overlap with the exopher periphery than the nearby neuronal soma periphery (*Figure 2C and D*; *Lemmon et al., 1995*). PI(4,5)P2 is typically enriched on the phagocytic cup and is removed rapidly upon phagosome sealing (*Botelho et al., 2000*). Our quantitative imaging indicates that most intact exophers undergo phagocytosis by the neighboring hypodermal membrane (*Figure 2A–D*).

We examined hypodermal actin association with exophers more closely in two ways. First, we divided exophers into three categories, early (stage 1), fully formed (stage 2), and exophers that had moved through the hypodermis to sites distant from the soma (stage 3) (*Figure 1—figure supplement 1*). Exophers had a similar incidence of F-actin positivity in all stages; 71% stage 1 (N = 7), 78% stage 2 (N = 14), and 66% stage 3 (N = 6) exophers were F-actin positive. Second, we captured early budding events via time-lapse imaging, which is challenging since exopher events are rare and their time of occurrence is random within the first three days of adulthood. We were successful in capturing six exopher budding events via time-lapse imaging (*Figure 2E*, *Video 2*). In 5/6 cases, hypodermal F-actin was not yet apparent when the neuronal bud first started to emerge, but in all cases F-actin was eventually acquired, indicating hypodermal recognition and engulfment.

If actin polymerization is important for phagocytosis of exophers, we would expect depletion of actin and/or the actin polymerization machinery to hinder degradation of exophers. Indeed, we found that depletion of hypodermal actin using tissue-specific RNAi against actins *act-1*, *act-2*, or *act-3* resulted in accumulation of exophers derived from the ALMR neuron, suggesting a failure in hypodermal phagocytosis (*Figure 2F*). Hypodermis-specific depletion of *arx-2*, encoding the Arp2 subunit of the Arp2/3 complex required for formation of branched actin during phagocytosis (*Goley and Welch, 2006*), also resulted in the accumulation of exophers and a reduction in starry night vesicles that derive from exopher degradation (*Figure 2G*). These experiments do not specify which actin isoforms are involved as actin RNAi reactions are likely to cross-react with one-another, and actin or arp2/3 perturbations are difficult to interpret in isolation, as they can affect many processes. Still, taken together, our data support the interpretation that a canonical actin-dependent hypodermal phagocytosis response is initiated in response to extruded exophers; hypodermis-specific disruption of actin

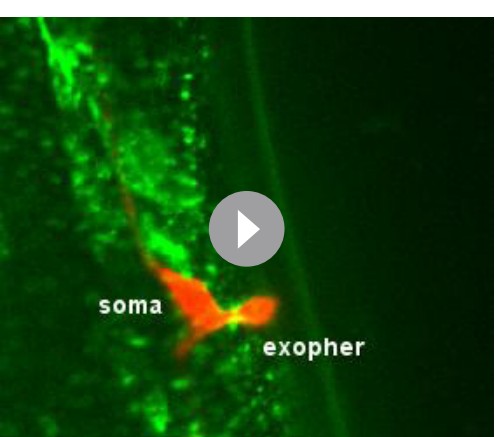

**Video 2.** Hypodermal F-actin dynamics mark engulfment of the neuronal exopher. An mCherry-filled ALMR neuron soma is shown during exopher production, with hypodermis-specific expression of mNeonGreen::UtrCH marking F-actin accumulation and dynamics during engulfment. Video depicts a 90 min time course, 15 min per frame. Scale Bar = 3 µm.
https://elifesciences.org/articles/82227/figures#video2

and key actin regulator Arp2/3 results in a block in hypodermal digestion that processes ALM-derived exophers (*Figure 2H*).

Once phagocytosis is complete, we would expect hypodermal endosome and lysosome markers to accumulate on exopher-phagosomes as phagocyte organelles fuse with the phagosome to promote phagosome maturation and the degradation of its contents. We found that hypodermally expressed mNeonGreen markers for distinct steps of phagosomal maturation label the periphery of intact ALMR neuron-derived exophers, but this labeling was weak, further supporting the interpretation that large exophers are usually associated with early stages of hypodermal phagocytosis (*Figure 3A–H*). In particular, we noted frequent weak labeling for early phagosome marker RAB-5 (42%, N = 29), late phagosome marker RAB-7 (70%, N = 24), and autophagosome marker LGG-1/LC3, suggesting that hypodermal early endosomes, late endosomes, and autophagosomes begin to associate with large phagocytosed exophers (*Figure 3A–F*). Low-level association with the exopher phagosome periphery may indicate transitory endosome and autophagosome association rather than full fusion at this stage. Notably, however, early and late phagosome marker labeling, and autophagosome marker labeling, is much more pronounced later, after fragmentation of the exopher-laden phagosome (*Figure 4A*). We observed little acquisition of hypodermally expressed lysosome marker LMP-1/LAMP by intact exopher-phagosomes, suggesting little lysosome association at the large single phagosome exopher stage (*Figure 3G and H*).

## Starry night vesicles represent maturing hypodermal phagosomes containing exopher-derived cargo

As noted above, exopher-laden phagosomes begin to break up into smaller 'starry night' vesicles about 2 hr after budding from a neuron. Phagosomes can break up into smaller vesicles during maturation, a process referred to as phagosome resolution (*Levin et al., 2016*; *Ghose and Wehman, 2021*). If these starry night vesicles represent a canonical phagosome resolution process, we would expect colocalization with phagosome maturation markers but not markers for other organelles. To test this hypothesis, we used confocal microscopy to measure percent colocalization of mCherry-labeled starry night vesicles with hypodermis-expressed markers for sequential steps of phagosome maturation, and other compartment markers. As expected for phagosomes, we found very little colocalization of starry night vesicles with Golgi marker AMAN-2 or basolateral recycling endosome marker RME-1 (*Figure 4A and B*). In contrast, our quantitative imaging documented strong colocalization of starry night vesicles with hypodermally expressed 2XFYVE(HRS), a biosensor for the early phagosomal lipid PI(3)P; RAB-10, a recycling regulator associated with early phagosomes and endosomes; and RAB-7, a marker of late phagosomes and endosomes (*Figure 4A and B*; *Levin et al., 2016*; *Lee et al., 2020*). RAB-10 association was particularly pronounced and may indicate recruitment of a RAB-5 GAP to promote maturation via the Rab5/Rab7 transition, kinesin recruitment to promote fragmentation and movement, and/or active recycling of some components out of the vesicles at this stage (*Liu and Grant, 2015*; *Etoh and Fukuda, 2019*; *Zajac and Horne-Badovinac, 2022*). We also found strong colocalization with LGG-1/LC3, a canonical marker of autophagosomes, recently found to contribute to phagosomal cargo degradation in *C. elegans* (*Figure 4A and B*; *Peña-Ramos et al., 2022*). We conclude that starry night vesicles reflect exopher-laden early and late phagosomes that have undergone vesiculation and suggest that markers found on starry night vesicles indicate a link between exopher phagosome resolution and phagosome maturation (*Figure 4D*; *Levin et al., 2016*; *Ghose and Wehman, 2021*).

We also examined the ultrastructure of membranes near the ALMR neuron in specimens initially visually selected for recent mCherry-positive ALMR exopher production. By electron microscopy, we identified multilamellar vesicles within the hypodermis near the ALMR neuron. These vesicles appeared in exopher-producing animals but not control animals, appearing similar to phagosome vesiculation products. The vesicles measured 200–500 nm in size, significantly smaller than the average extruded large vesicle exopher (~3 μm). These results are consistent with a model in which starry night vesicles represent phagosomes undergoing resolution processing (*Figure 4C*).

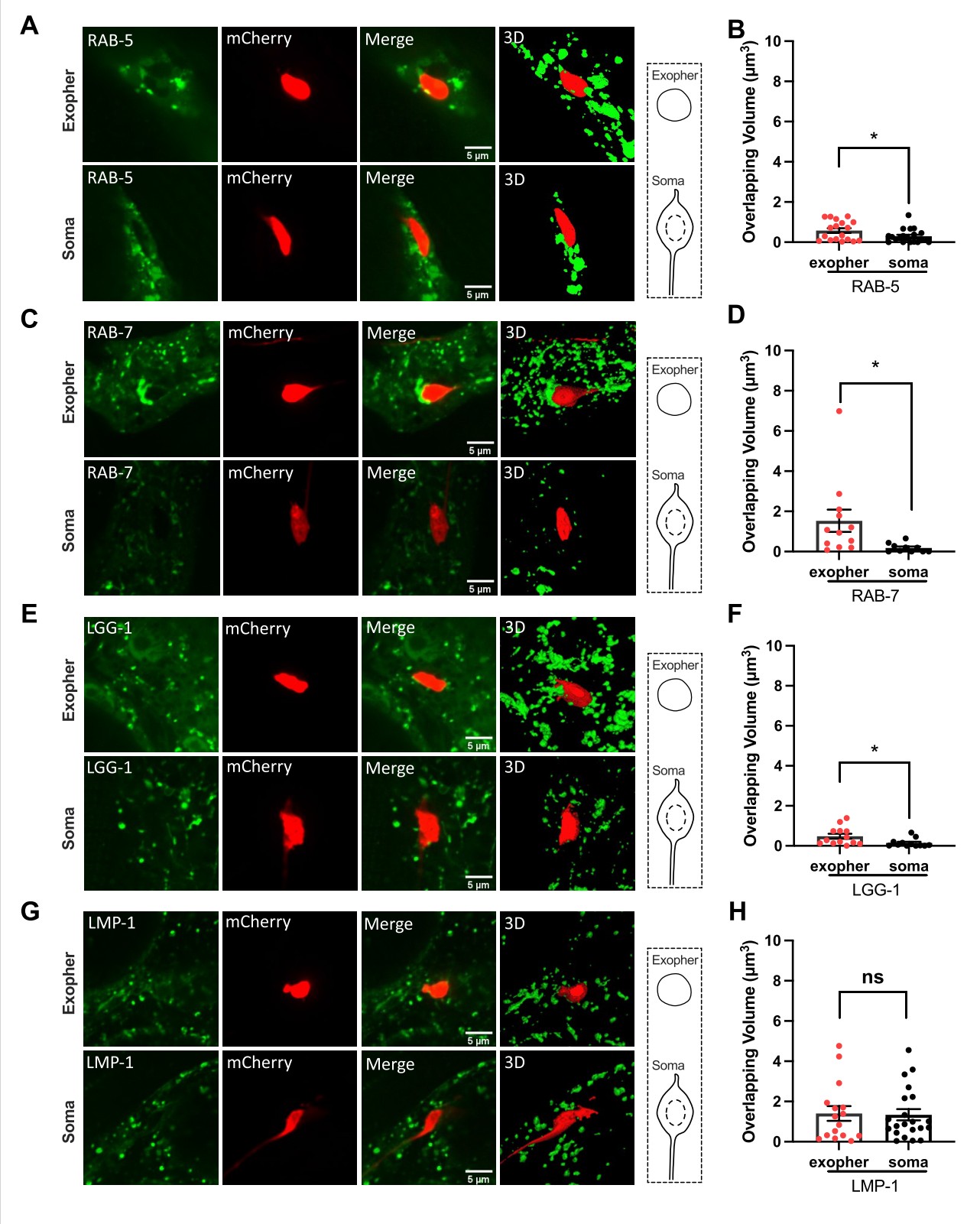

**Figure 3.** Hypodermal endosomes and autophagosomes, but not lysosomes, begin recruitment prior to exopher-laden phagosome vesiculation. (**A, C, E, G**) Fluorescence micrographs are shown for an mCherry-labeled ALMR neuron-derived exopher, or the similarly sized mCherry-labeled ALMR neuronal soma from the same neuron. Surrounding hypodermis-specific expression of (**A**) mNeonGreen::RAB-5 (an early endosome marker), (**C**) mNeonGreen::RAB-7 (a late endosome marker), (**E**) mNeonGreen::LGG-1/LC3 (an autophagosome marker), and (**G**) LMP-1::mNeonGreen (a lysosome

*Figure 3 continued on next page*

*Figure 3 continued*

marker), driven by the hypodermis-specific *semo-1* promoter, are shown. A merged image and a merged 3-D projection are shown for each example. Scale bar, 5 μm. (**B, D, F, H**) As a measure of hypodermal marker recruitment to the exopher, we measured the volume of overlapping signal between the hypodermal marker and ALMR-neuron derived exopher, comparing to the ALMR neuronal soma as a control, *p<0.05 by two-tailed unpaired *t*-test. N = 13 (LGG-1), N = 20 (RAB-5), N = 12 (RAB-7), and N = 21 (LMP-1).

The online version of this article includes the following source data for figure 3:

**Source data 1.** Numerical data for *Figure 3*.

## Exopher content degradation requires phagosome maturation and lysosomal activity

Unexpectedly, we measured low colocalization of exopher-derived phagosomal vesicles in the hypodermis with lysosome marker LMP-1, whereas phagosome maturation should require lysosome fusion as a late step. We hypothesized that fusion of exopher-laden phagosomes with hypodermal lysosomes might rapidly degrade the neuron-derived mCherry found in exophers, making it difficult to capture instances of exopher-derived mCherry in late-stage phagolysosomes. To test this idea, we assayed the effects of a *cup-5*/mucolipin mutant, defective in lysosome function, on exopher and starry night. Consistent with this hypothesis, in *cup-5* mutants colocalization with the LMP-1 lysosome marker was greatly increased, the size of starry night vesicles was significantly larger, and the average fluorescence intensity of mCherry in starry night vesicles was significantly higher (*Figure 5A–D*). Our results suggest that most exopher-derived material taken up by the hypodermis from the neuron is degraded in hypodermal phagolysosomes, and that lysosome fusion is a late step in exopher processing. Lysosome function was not required for the transition from large exopher-containing phagosome into starry night phagosome fragments.

To better define the connection of exopher-phagosome maturation, fragmentation, and degradation, we also examined the effects of loss of *sand-1* using a temperature-sensitive allele. SAND-1/Mon1 is part of a RAB-7 exchange factor complex required for RAB-5 to RAB-7 conversion during early to late phagosome maturation. Interestingly, we found that loss of *sand-1* caused a significant increase in the detection of intact exopher-laden phagosomes, along with loss of most smaller starry night fragmented phagosomes (*Figure 5E and F*). This processing-stalled phenotype was already apparent on day 1 of adulthood and persisted at least through day 4 of adulthood, the time by which nearly all WT exopher-phagosomes have normally converted to starry night (*Figure 5F*). We conclude that conversion from a large exopher-laden phagosome to smaller starry night phagosomal fragments is important for cargo degradation, and that the molecular changes associated with early to late phagosome maturation, such as loss of RAB-5 and acquisition of GTP-bound RAB-7, are required for fragmentation into starry night vesicles.

To further probe requirements for phagosome fragmentation, we tested *arl-8*, which encodes a small GTPase associated with lysosomes and endosomes that has been reported to affect phagosome or lysosome vesiculation by recruiting kinesins (*Fazeli et al., 2018*; *Levin-Konigsberg et al., 2019*; *Fazeli et al., 2023*). We found that *arl-8* hypomorphic mutants exhibit a defect in transitioning from large exopher-laden phagosomes to smaller starry night phagosomal vesicles. Although less pronounced than in *sand-1* mutants, perhaps due to residual ARL-8 activity, we measure a significant difference in persistence of large vesicle exophers at days 3–4 of adulthood in *arl-8* (*Figure 5—figure supplement 1*). Because the ARL-8 GTPase is thought to recruit kinesins to endolysosomal compartments, our observations suggest a role for cytoskeletal motors in exopher-laden phagosome vesiculation and support a model that links vesiculation of exopher phagosomes to exopher content degradation.

## Hypodermal CNT-1 and RAB-35 are required for exopher maturation

Given our observation of hypodermal PI(4,5)P2 enrichment around early exophers (*Figure 2C and D*), we also analyzed the effects of mutants thought to influence PI(4,5)P2 levels on phagosomes. ARF-GAP CNT-1 and small GTPase RAB-35 function in some cell corpse engulfment events (*Haley et al., 2018*; *Kutscher et al., 2018*). CNT-1 and RAB-35 have been proposed to act in the removal of PI(4,5)P2 during phagosome maturation, via deactivation of another small GTPase, ARF-6, a known activator of PI-kinases (*Kutscher et al., 2018*). Our own previous work showed a requirement in *C. elegans* for

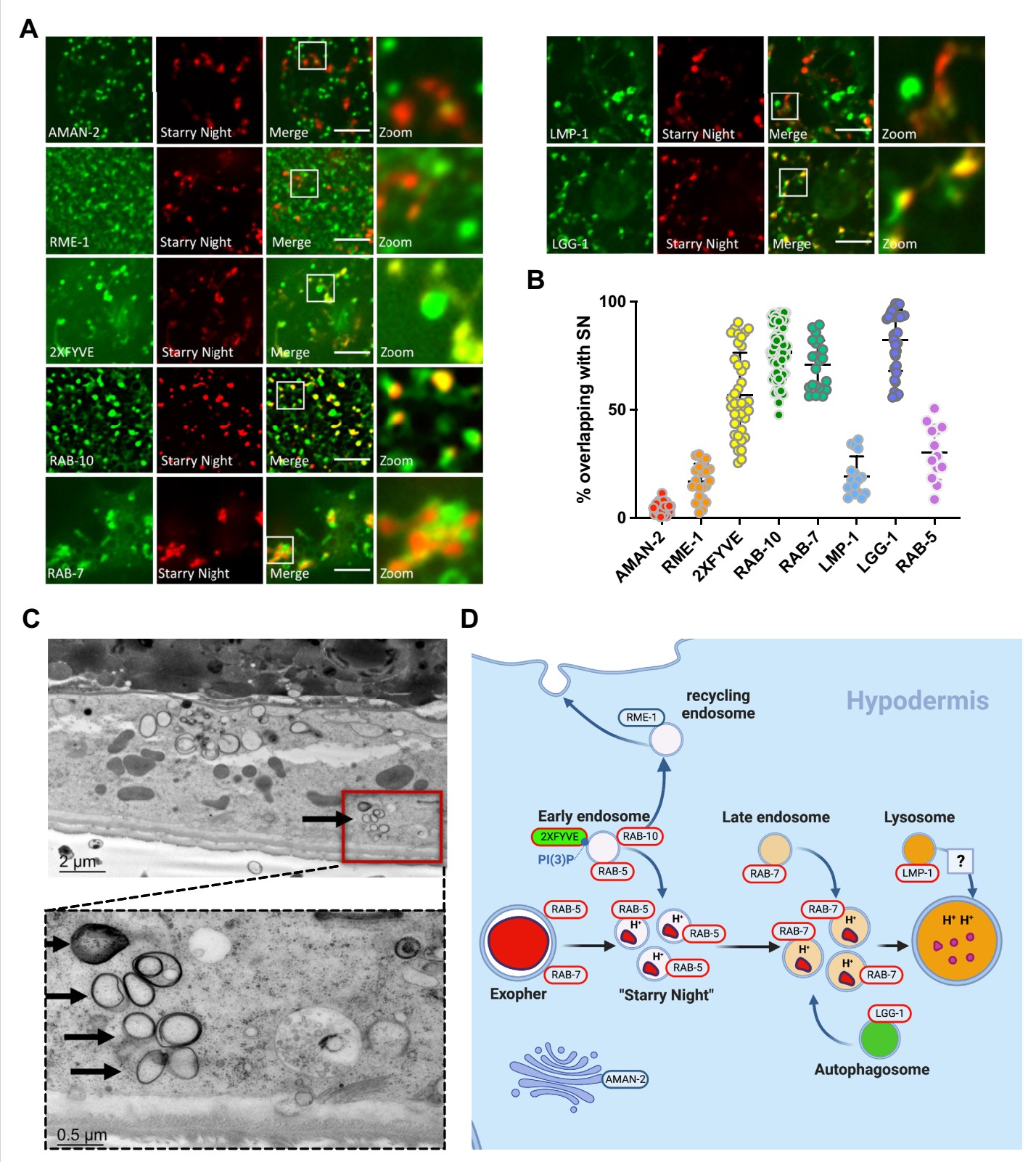

**Figure 4.** Starry night vesicles are exopher-phagosome vesiculation products that further fuse with hypodermal endosomes. (**A**) Fluorescence micrographs show ALMR-exopher-derived mCherry labeling exopher-laden phagosome vesiculation products. Hypodermis-specific expression of AMAN-2::mNeonGreen (a Golgi marker), mNeonGreen::RME-1 (a recycling endosome marker), mNeonGreen::2XFYVE (an early endosome/phagosome marker and PI(3)P biosensor), mNeonGreen::RAB-10 (an early phagosome/endosome marker and recycling regulator), mNeonGreen::RAB-7 (a late

*Figure 4 continued on next page*

*Figure 4 continued*

endosome/phagosome marker), LMP-1::mNeonGreen (a lysosome marker), and mNeonGreen::LGG-1/LC3 (an autophagosome marker), each driven by the hypodermis-specific *semo-1* promoter, are shown. Scale bar, 5 µm. (**B**) Graph quantifying colocalization of each marker with starry night exopher-phagosome vesiculation products. AMAN-2, N = 22; RME-1, N = 20; 2XFYVE; N = 28; RAB-10, N = 30; RAB-7, N = 21; LMP-1 N = 17; LGG-1, N = 22, RAB-5, N = 12. (**C**) Thin section electron micrograph showing candidate phagosome-derived vesicles (arrows) within the hypodermis near the ALMR soma. (**D**) Diagram summarizing interpretation of events, including early endosome, late endosome, and autophagosome fusion with exopher-laden phagosome vesiculation products.

The online version of this article includes the following source data for figure 4:

**Source data 1.** Numerical data for *Figure 4*.

CNT-1 and ARF-6 in endosomal regulation in the intestinal epithelium (*Sato et al., 2008*). Therefore, we tested *cnt-1, rab-35,* and *arf-6* for roles in exopher-associated phagocytosis.

We found that *cnt-1* and *rab-35* mutants accumulate large intact exophers, with higher levels of exophers present until at least day 4 of adulthood in *cnt-1* mutants, by which point nearly all WT exophers are normally consumed (*Figure 6D, G, and H*; *Figure 6—figure supplement 1*). Importantly, we find that hypodermis-specific expression of CNT-1::mNG or mNG::RAB-35, but not touch-neuron-specific expression, rescued exopher and starry night numbers in their respective mutants (*Figure 6H and I*). This tissue-specific rescue indicates that CNT-1 and RAB-35 function in the hypodermis for exopher clearance (*Figure 6M*). Hypodermis-specific mNG::RAB-35 could be clearly visualized labeling the periphery of large exophers (N = 12), but not around the neuronal soma, as predicted for a direct role in exopher clearance (*Figure 6B*). Hypodermis-specific CNT-1::GFP also often labeled exophers more than somas (N = 22), but was found in characteristic puncta at the exopher periphery rather than labeling the periphery smoothly like RAB-35 (*Figure 6A*). Taken together, our results support a direct role for RAB-35 and CNT-1 in regulating hypodermal phagosome formation during neuronal exopher processing.

## ARF-6 acts in the hypodermis to influence exopher production in the neuron

Given the proposed role of CNT-1 and RAB-35 in ARF-6 downregulation, we directly tested the role of ARF-6 in exopher processing using a null allele *arf-6(tm1447)*, in which nearly the whole *arf-6* gene is deleted. In contrast to our results with *cnt-1* and *rab-35*, where exophers accumulate, we found a significant reduction in exopher numbers in *arf-6* mutants in ALMR neurons expressing mCherry or GFP (*Figure 6D, E, and J*). Furthermore, *arf-6* mutants lacked most starry night, indicating that the loss of exophers was not due to an increased rate of hypodermal processing (*Figure 6J*). To confirm these results, we used a *daf-2* mutant background to expand the dynamic range of our assay since *daf-2* mutation greatly increases the basal level of exopher production (*Cooper et al., 2021*). This experiment confirmed a pronounced loss of exophers and starry night phagosomes in *arf-6* mutants (*Figure 6K*).

To investigate the effect of *arf-6* deficiency in more detail, we captured time-lapse video of neuronal somata, comparing *daf-2* mutants with *daf-2; arf-6* double mutants. Although we were not able to capture enough full exophergenesis events, among more than 50 ~4 hr time-lapse videos of each genotype to analyze the effects of *arf-6* on exophergenesis, we found similar rates of small bud production in both genotypes, suggesting that very early events in exophergenesis in *arf-6* mutants occur at normal frequency (*Figure 6—figure supplement 4*; *Videos 3 and 4*). Our results indicate that ARF-6 is important for exopher production, likely affecting small bud growth to form the large exopher bud, or to prevent regression of exopher buds once formed.

Importantly, re-expression of *arf-6(+)* in just the hypodermis in the *arf-6(Δ)* background rescued exopher and starry night numbers (*Figure 6J*); and *arf-6* depletion via hypodermis-specific RNAi conferred the same reduction in exopher and starry night numbers as the *arf-6(Δ)* mutants (*Figure 7A*). Both lines of testing support the interpretation that ARF-6 activity in the hypodermis facilitates the successful production of exophers in the neuron. We also measured a higher level of hypodermal ARF-6::mNeonGreen puncta associated with the exopher periphery than around the neuronal soma, supporting the idea of a direct role for hypodermal ARF-6 in promoting neuronal exopher production (*Figure 6C*).

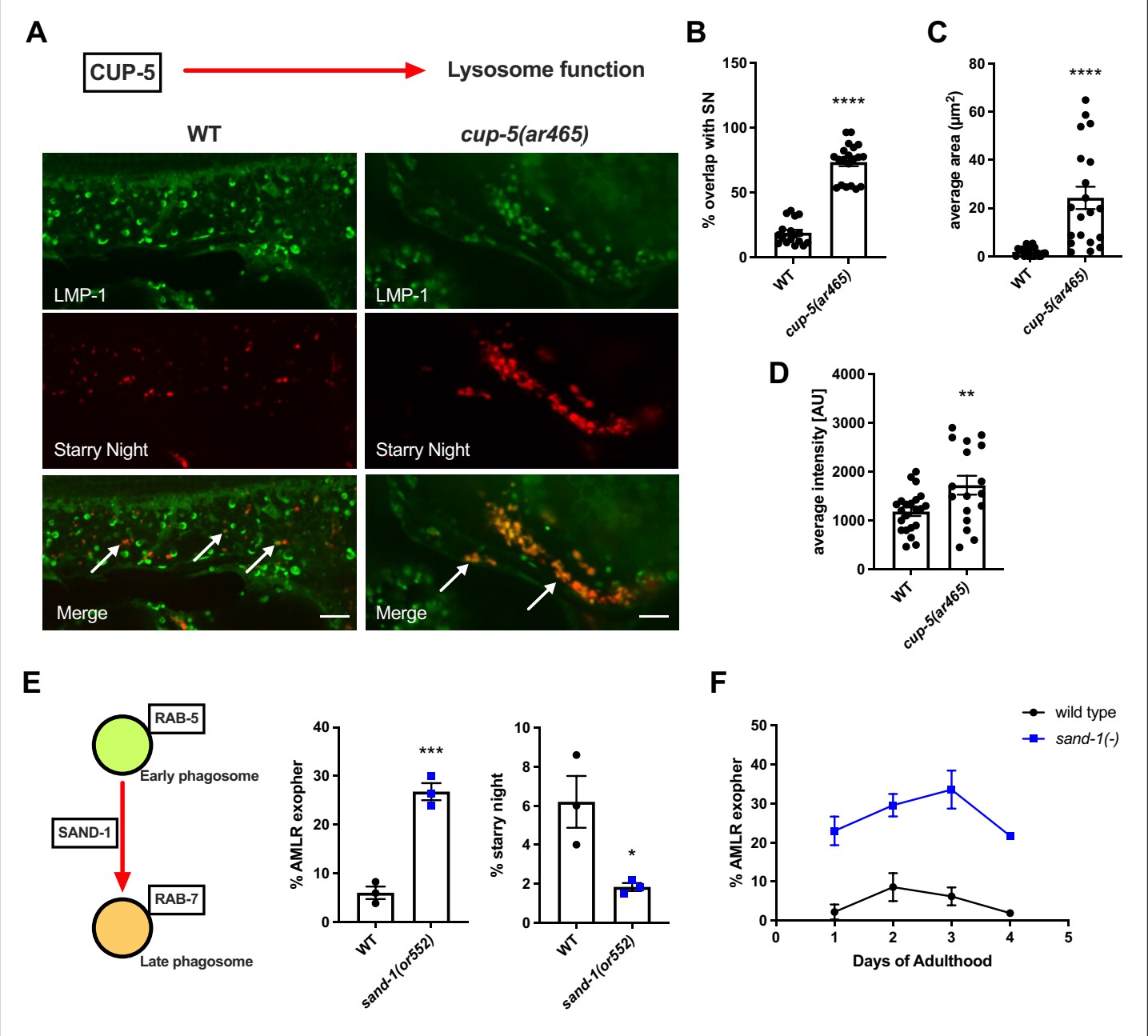

**Figure 5.** Lysosome function is required for the end-stage degradation of phagocytosed exopher cargo. (**A**) Fluorescence micrographs show ALMR-exopher-derived mCherry labeling exopher-phagosome vesiculation products (starry night) in red, with hypodermis-specific expression of LMP-1::mNeonGreen (a lysosome marker) shown in green. Wild-type and *cup-5(ar465)* mutant images are shown. Arrows indicate LMP-1:mNG that colocalized with neuron-derived mCherry vesicles. Scale bar, 5 µm. (**B**) Histogram quantifying colocalization of the LMP-1 lysosome marker with ALMR-exopher-derived vesicles in wild-type and *cup-5* mutant. (**C**) Histogram quantifying the average size of ALMR-exopher-derived vesicles in wild-type and *cup-5* mutant. (**D**) Histogram quantifying average the florescence intensity of mCherry signal in ALMR-exopher-derived vesicles in wild-type and *cup-5* mutant. (**B–D**) N = 17 wild-type, N = 20 *cup-5(ar465)* ****p<0.0001, **p<0.01 by two-tailed unpaired *t*-test. (**E**) Histogram quantifying exopher and starry night numbers in wild-type and *sand-1(or552)* mutant **** p<0.0001, *p<0.05 by the Cochran–Mantel Haenszel (CMH) test, N = 50 per trial over three trials. (**F**) Line graph quantifying exopher numbers at different days of adulthood in wild-type and *sand-1(or552)* mutant. N = 50 per trial over three trials.

The online version of this article includes the following source data and figure supplement(s) for figure 5:

**Source data 1.** Numerical data for *Figure 5*.

**Figure supplement 1.** ARL-8 promotes the vesiculation of exophers-laden phagosomes.

**Figure supplement 1—source data 1.** Numerical data for *Figure 5—figure supplement 1*.

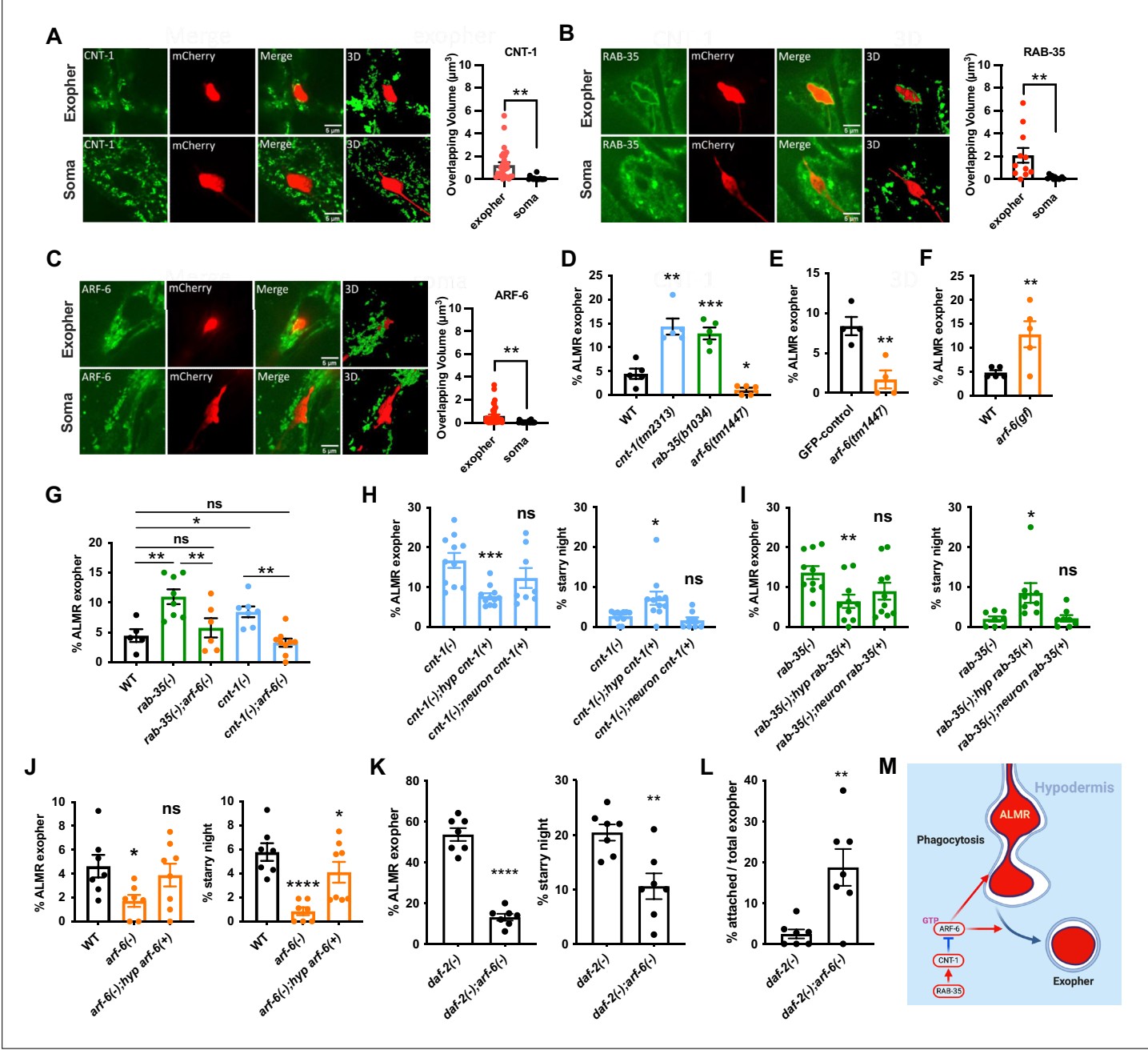

**Figure 6.** Hypodermal ARF-6 influences neuronal exopher production and interacts with CNT-1 and RAB-35 to complete hypodermal phagocytosis. (A–C) Fluorescence micrographs showing hypodermally expressed CNT-1/ARF-GAP, GTPase RAB-35, and GTPase ARF-6 associated with the periphery of intact exophers. Histograms indicate overlapping volume of hypodermal marker with exopher and neuronal soma periphery, **p<0.01 by two-tailed unpaired *t*-test. CNT-1 N = 22; RAB-35 N = 12; ARF-6 N = 24. Scale bar, 5 µm. (D) Histogram depicting exopher frequency in wild-type, *cnt-1(tm2313)*, *rab-35(b1034)*, and *arf-6(tm1447)* mutants. ***p<0.001, **p<0.01, *p<0.05 by the Cochran–Mantel Haenszel (CMH) test, n = 50 over five trials. (E) Histogram depicting exopher frequency in animals expressing touch neuron-specific GFP from the *mec-17* promoter (*uIs31*), compared with *uIs31; arf-6(tm1447)* animals. **p<0.01 by the CMH test, N = 50 per trial over four trials. (F) Histogram depicting exopher frequency in wild-type and *arf-6(ns388)* gain-of-function mutant. **p<0.01 by the CMH test, N = 50 per trial over five trials. (G) Histogram depicting exopher frequency in wild-type, *cnt-1* and *rab-35* single mutants and double mutants with *arf-6(tm1447)*. **p<0.01, *P<0.05 by the CMH test, N = 50 per trial over five trials. (H–J) Rescue experiments showing exopher and starry night frequency rescue by hypodermis-specific expression of *rab-35*, *cnt-1*, and *arf-6* in their cognate mutants. (K) Histogram depicting exopher and starry night numbers in *daf-2* mutant and in the double mutant with *daf-2* and *arf-6*. ****p<0.0001, **p<0.01 by the CMH test, N = 50, over seven trials. (L) Histogram depicting attached exopher rate in *daf-2(e1370)* mutant background to potentiate baseline exophergenesis and double mutant with *arf-6*. **p<0.01 by two-tailed unpaired *t*-test. N = 50 per trial over seven trials. ****p<0.0001, ***p<0.001,

*Figure 6 continued on next page*

*Figure 6 continued*

\*\*p<0.01, \*p<0.05 by the CMH test, N = 50 per trial. (**M**) Diagram indicating hypodermal ARF-6 influence on exopher budding and engulfment, and negative regulation of ARF-6 by CNT-1 and RAB-35.

The online version of this article includes the following source data and figure supplement(s) for figure 6:

**Source data 1.** Numerical data for *Figure 6*.

**Figure supplement 1.** CNT-1 is required for exopher-laden phagosome maturation.

**Figure supplement 1—source data 1.** Numerical data for *Figure 6—figure supplement 1*.

**Figure supplement 2.** Hypodermal overexpression of CNT-1 can suppress exopher accumulation due to loss of RAB-35, but not vice versa.

**Figure supplement 2—source data 1.** Numerical data for *Figure 6—figure supplement 2*.

**Figure supplement 3.** Hypodermal ARF-6 puncta decorate the periphery of engulfed exophers and are increased upon loss of CNT-1.

**Figure supplement 3—source data 1.** Numerical data for *Figure 6—figure supplement 3*.

**Figure supplement 4.** *arf-6* mutants maintain early ALM budding.

**Figure supplement 4—source data 1.** Numerical data for *Figure 6—figure supplement 4*.

## ARF-6 functions downstream of RAB-35 and CNT-1

We further tested the relationship of ARF-6 to CNT-1 and RAB-35 via genetic epistasis. We found that the elevated number of exophers found in *cnt-1* and *rab-35* mutants was suppressed in *cnt-1;arf-6* and *rab-35;arf-6* double mutants (*Figure 6G*). This is consistent with RAB-35 and CNT-1 acting as negative regulators of ARF-6. Supporting this interpretation, we found that an *arf-6(ns388)* gain-of-function mutant (*Kutscher et al., 2018*) produced increased exopher numbers, similar to *rab-35* and *cnt-1* mutants, and the opposite of the *arf-6* loss-of-function mutants (*Figure 6F*).

Furthermore, we found that *rab-35* mutant exopher accumulation could be partly rescued by hypodermal CNT-1 overexpression, while *cnt-1* mutant exopher accumulation was not rescued by RAB-35 overexpression (*Figure 6—figure supplement 2*). We also found greater association of ARF-6::mNeonGreen with exophers in *cnt-1* mutants than wild-type controls (*Figure 6—figure supplement 3*). Taken together, these results are consistent with a model in which RAB-35 upregulates CNT-1, and CNT-1 downregulates ARF-6 via ARF-GAP activity within the hypodermal cell (*Figure 6M*).

## SEC-10/exocyst and PPK-1 PI-5 kinase may act with ARF-6

To identify relevant ARF-6 effectors participating in exopher production and phagocytosis, we performed hypodermis-specific RNAi knockdown of *C. elegans* homologs of known mammalian Arf6 effectors (*Figure 7A–F*). Among these, we found that only knockdown of *sec-10*, a component of

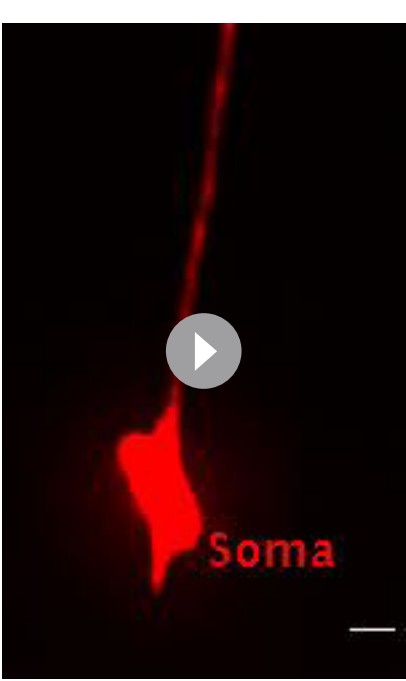

**Video 3.** Exopher budding in a *daf-2* mutant. An mCherry-filled ALMR neuron soma in a *daf-2(e1370)* mutant is shown during exopher production. Video depicts a 4 hr time course, 4 min per frame. Scale Bar = 3 μm.

https://elifesciences.org/articles/82227/figures#video3

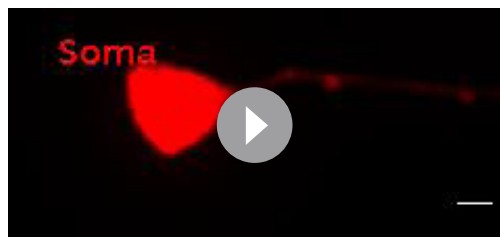

**Video 4.** Early bud formation in *daf-2; arf-6* double mutant. An mCherry-filled ALMR neuron soma in a *daf-2(e1370); arf-6(tm1447)* double mutant is shown producing a small bud from the soma. Video depicts a 4 hr time course, 4 min per frame. Scale Bar = 3 μm.

https://elifesciences.org/articles/82227/figures#video4

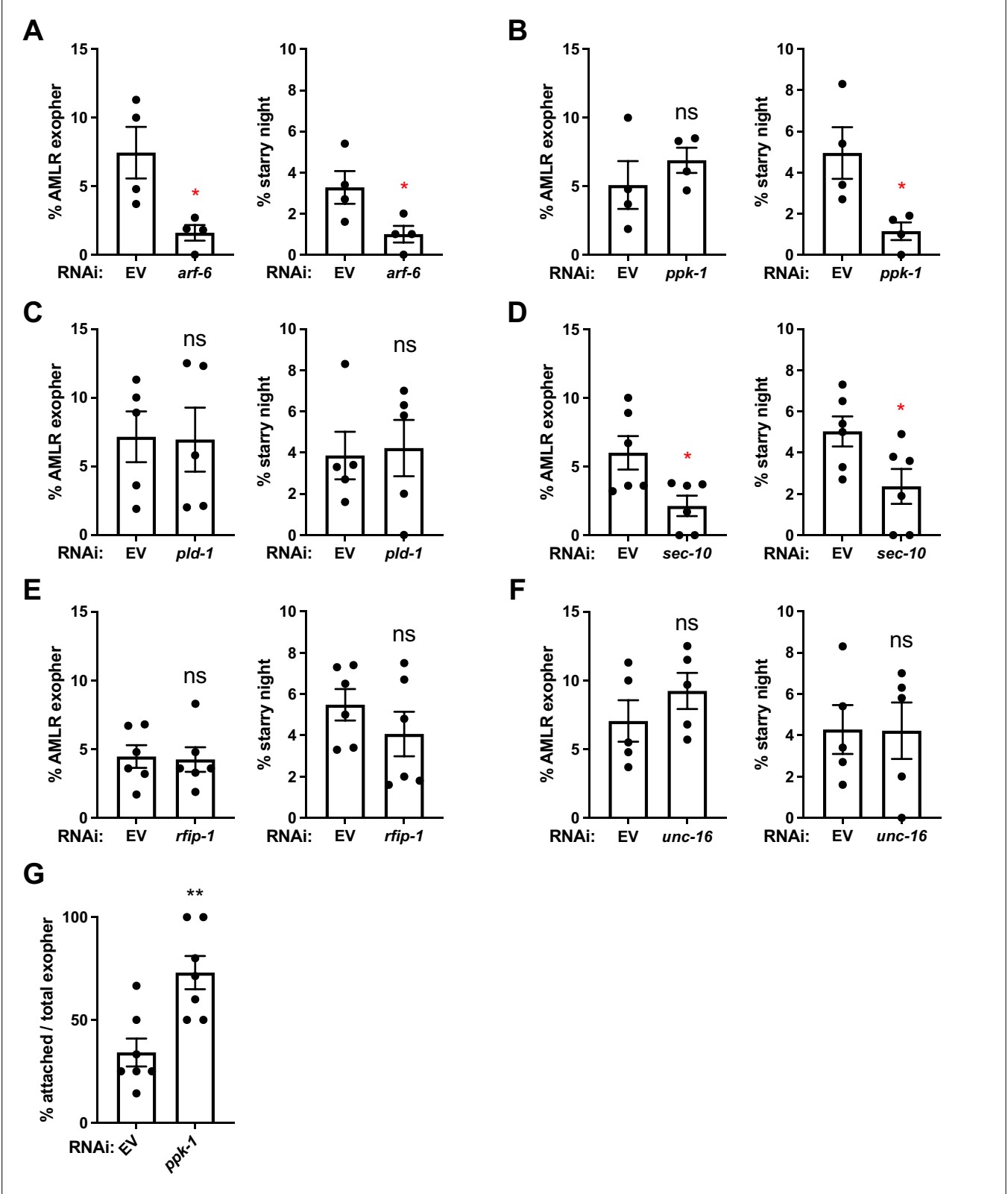

**Figure 7.** Hypodermis-specific RNAi identifies SEC-10/exocyst and PPK-1/PI-5 kinase as likely ARF-6 effectors in exopher production and phagocytosis. (**A–F**) Histograms depict exopher and starry night numbers after hypodermis-specific RNAi for candidate ARF-6 effectors. EV indicates empty vector control, *arf-6* encodes GTPase ARF-6/Arf6, *ppk-1* encodes phosphatidylinositol-4-phosphate 5-kinase, *rfip-1* encodes the *C. elegans* homolog of Rab11-FIP3/Arfophilin, *pld-1* encodes the *C. elegans* homolog of phospholipases D1 and D2, *unc-16* encodes the *C. elegans* homolog of C-Jun-amino-

*Figure 7 continued on next page*

*Figure 7 continued*

terminal kinase-interacting protein 3 (JIP3), and *sec-10* encodes the *C. elegans* homolog of Exocyst subunit Sec10. **p<0.01, *p<0.05 by the Cochran–Mantel Haenszel (CMH) test, N = 50 per trial over four trials. (**G**) Histogram depicting attached exopher rate in hypodermis-specific RNAi for *ppk-1*. **p<0.01 by two-tailed unpaired *t*-test. N = 50 per trial over seven trials.

The online version of this article includes the following source data for figure 7:

**Source data 1.** Numerical data for *Figure 7*.

the vesicle tethering complex Exocyst, produced an overall phenotype similar to *arf-6*, with significantly reduced exopher and starry night frequency (*Figure 7D*). Our data suggest that ARF-6 and SEC-10 function together in the hypodermis in a process required for the neuron to efficiently produce exophers. Given the nature of SEC-10 in vesicle tethering and fusion with the plasma membrane, and the proposed role of mammalian Arf6 in membrane delivery during phagocytosis (*Niedergang et al., 2003*), ARF-6-dependent delivery of membrane to the phagocytic cup via recycling vesicles may be the key process affected by SEC-10.

We found a different effect for hypodermal disruption of another candidate ARF-6 effector *ppk-1*, encoding a phosphatidylinositol 4-phosphate 5-kinase that can convert PI(4)P to PI(4,5)P2 on the plasma membrane (*Doughman et al., 2003*). Hypodermis-specific RNAi of *ppk-1* reduced starry night frequency, but did not reduce the number of exophers, which is different from *arf-6* and *sec-10* phenotypes (*Figure 7B*). We also noted that hypodermis-specific depletion of *ppk-1* produced higher than normal numbers of exophers remaining attached to the neuronal soma and measured a similar defect in *arf-6* mutants for the few exophers that are produced without ARF-6 (*Figure 6L* and *Figure 7G*). Given these results, and data on PPK-1 homologs as Arf6 effectors in other organisms, we propose that PPK-1 participates in the exopher engulfment process downstream of ARF-6, but only in the later phase required for phagosome release (sealing), since *ppk-1* knockdown only phenocopies the later part of the *arf-6* mutant phenotype (*Figure 6L* and *Figure 7A, D*). Our data may indicate dual roles for ARF-6, one early in exopher recognition/interaction by the hypodermis, and another role later in hypodermal phagosome completion.

## *ced-1*, *ttr-52*, and *anoh-1* mutants, but not *ced-10* mutants, reduce exopher production

As the reduction in exopher production found in *arf-6* mutants was surprising, we sought to compare these results with other mutants that might affect exopher phagocytosis by the hypodermis. To test this directly, we analyzed effects of loss of the CED-1/DRAPER/MEGF10 phagocytic receptor, as well as CED-10/Rac, a master regulator of actin dynamics and membrane deformation during phagocytosis that acts in a parallel pathway to CED-1 (*Mangahas and Zhou, 2005*). CED-1 and CED-10 are well known for their roles in phagocytosis and degradation of apoptotic cell corpses in *C. elegans* and other organisms, but how such pathways would interact with exophers budding from living neurons was unclear. In the case of apoptotic cells, loss of CED-1 or CED-10 leads to an accumulation of cell corpses (*Zhou et al., 2001*). We found that *ced-1* mutants had significantly reduced exopher and starry night numbers, similar to the effects we observed in *arf-6* mutants and *sec-10* hypodermis-specific knockdown (*Figure 8A*). Furthermore, *ced-1* mutant exopher production was rescued by hypodermis-specific re-expression of CED-1, indicating a hypodermal focus for these CED-1 effects (*Figure 8B*).

CED-1 recognition of phagocytic targets usually depends upon surface exposure of phosphatidylserine (PS) and transthyretin-like protein TTR-52 that helps link the CED-1 extracellular domain to exposed PS on target cells (*Wang et al., 2010*). ANOH-1, the *C. elegans* homolog of Ca$^{2+}$-activated phospholipid scramblase TMEM16F, is required for PS-exposure and CED-1-mediated recognition of necrotic cells (*Li et al., 2015*; *Furuta et al., 2021*). We found that *ttr-52* and *anoh-1* mutants display strongly reduced levels of exopher production similar to *ced-1* mutants (*Figure 8D*). Taken together, these results suggest that PS exposure contributes to exopher recognition and further suggest that a specific engulfing cell interaction with the neuron strongly influences the eventual production of the exopher itself.

To better determine whether the effects of CED-1 on exopher production might be via direct interaction of hypodermal CED-1 with the emerging exopher, we examined CED-1 protein association with exophers. In particular, we used a hypodermis-specific, GFP-tagged, version of CED-1

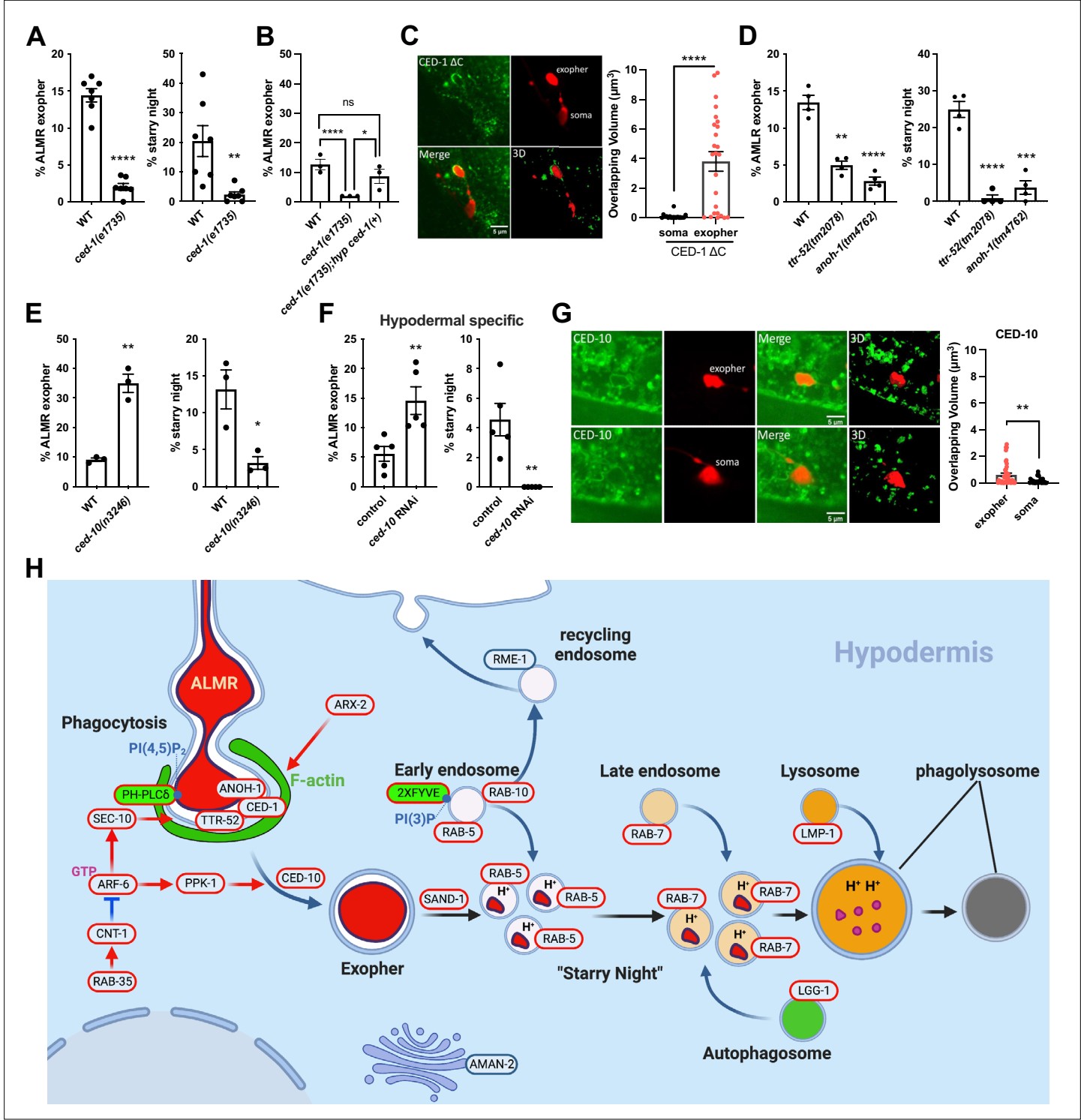

**Figure 8.** CED-1 functions in exopher recognition and exopher production. (**A**) Histogram depicting exopher and starry night numbers in wild-type and *ced-1(e1735)* mutant. N = 22. ****p<0.0001, **p<0.01 by the Cochran–Mantel Haenszel (CMH) test, N = 50 per trial over seven trials. (**B**) Histogram depicting exopher numbers in wild-type, *ced-1(e1735),* and *ced-1(e1735)* expressing CED-1(+) from a hypodermis-specific promoter. N = 22. ****p<0.0001, **p<0.01 by the CMH test, N = 50 per trial over three trials. (**C**) Fluorescence micrographs showing intact exophers interacting with hypodermal CED-1ΔC::GFP. Scale bar, 5 μm. Histogram depicts overlapping volume of hypodermal CED-1ΔC::GFP with exopher. ****p<0.0001 by two-tailed unpaired *t*-test. (**D**) Histogram depicting exopher and starry night numbers in wild-type, *ttr-52(tm2078)*, and *anoh-1(tm4762)* mutants. ****p<0.0001, ***p<0.001, **p<0.01 by the CMH test, N = 50 per trial over four trials. (**E**) Histogram depicting exopher and starry night numbers in wild-type and *ced-10(n3246)* mutants. ****p<0.0001, **p<0.01 by the CMH test, N = 50 per trial over three trials. (**F**) Histogram depicting exopher and starry

*Figure 8 continued on next page*

*Figure 8 continued*

night numbers after hypodermis-specific knockdown in empty vector control and *ced-10(RNAi)* animals. ****p<0.0001, **p<0.01 by the CMH test, N = 50 per trial over three trials. (**G**) Fluorescence micrographs showing intact exophers interacting with hypodermal mNeonGreen::CED-10. Scale bar, 5 μm. Histogram depicts overlapping volume of hypodermal mNeonGreen::CED-10 with exopher and soma. **p<0.01 by two-tailed unpaired *t*-test. (**H**) Model of ALMR-neuron derived exopher recognition, engulfment, and processing by the surrounding hypodermis.

The online version of this article includes the following source data and figure supplement(s) for figure 8:

**Source data 1.** Numerical data for *Figure 8*.

**Figure supplement 1.** Hypodermal CED-1 distinguishes between exopher and soma.

**Figure supplement 2.** *arf-6* and *ced-1* mutants display reduced touch sensitivity in old age.

**Figure supplement 2—source data 1.** Numerical data for *Figure 8—figure supplement 2*.

that lacks C-terminal CED-1 sequences, and increases the duration of CED-1/target associations that are normally otherwise quite transient (*Zhou et al., 2001*). Indeed, hypodermal CED-1ΔC::GFP clearly labeled the periphery of exophers undergoing engulfment. The pattern of association of CED-1ΔC::GFP with the exopher was striking in that CED-1ΔC::GFP became enriched at sites of early exopher budding, but did not label the still-attached neuronal soma (*Figure 8C*, *Figure 8—figure supplement 1*). Since the hypodermal CED-1ΔC::GFP we used is constitutively expressed, we attribute the exopher surrounding CED-1ΔC::GFP signal to CED-1 recruitment by exopher-surface signals.

Our results with CED-10/Rac were quite different than CED-1, TTR-52, and ANOH-1. We found that *ced-10* mutants, and hypodermis-specific *ced-10* RNAi animals, accumulate exophers and display strongly reduced frequency of starry night vesicles (*Figure 8E and F*). These results indicate that hypodermal CED-10 does not affect neuronal exopher generation like CED-1, ARF-6, or SEC-10, but strongly affects completion of exopher phagocytosis, a prerequisite for phagosome fragmentation into starry night vesicles. We also found that hypodermis-specific mNeonGreen::CED-10 is enriched on the hypodermal membrane surrounding the exopher compared to the hypodermal membrane surrounding the neuronal soma, suggesting a direct role for CED-10 in exopher-laden phagosome formation (*Figure 8G*).

We conclude that the conserved CED-1/DRAPER phagocytic receptor plays a role in recognition and removal of neuronally derived exophers, and along with ARF-6 and SEC-10, supports activities in the hypodermis that greatly influence neuronal exopher production. Other phagocytic regulators, such as CED-10/Rac, do not appear important for such non-autonomous regulation of exophergenesis, but are required to complete phagocytosis and remove the exopher once it has formed. Our results emphasize the existence of two classes of phagocytic regulators, those required to promote exopher production in the neuron, and those required only to execute phagocytosis.

## Discussion

The exopher is a recently identified giant extracellular vesicle that has the capacity to carry large amounts of cargo out the neuron, especially under high-stress conditions (*Melentijevic et al., 2017*; *Cooper et al., 2021*). Here we investigated the fate of mCherry-filled exophers released by the *C. elegans* ALMR touch neuron, finding that the surrounding hypodermis engulfs the exopher via defined phagocytosis steps, then processes the exopher-laden phagosome via phagosome maturation, ultimately leading to the degradation of most neuron-derived mCherry cargo in hypodermal phagolysosomes. The exopher response appears to be a mechanism to expel toxic cargo such as protein aggregates from the neuron, which is activated when other proteostasis pathways, such as the ubiquitin-proteasome and autophagy pathways, prove insufficient to the proteostatic load (*Melentijevic et al., 2017*). Exophergenesis is likely to be especially important in cells such as neurons that cannot dilute accumulated toxins by cell division. Uptake of toxic or excess material by another cell provides a fresh chance to degrade components that the neuron-intrinsic pathways could not handle. In the case of the ALMR touch neuron in *C. elegans*, the hypodermal receiving cell Hyp7 is a very large syncytium formed by the fusion of many cells during development. Hyp7 degradative capacity is expected to far outstrip that of the neuron expelling an exopher. The exopher, as a phagocytic cargo, presents an unusual challenge in that it is often as large as an apoptotic cell, but remains attached to a living neuron during early stages of engulfment (*Melentijevic et al., 2017*).

## Neuronal exopher production requires phagocyte recognition

Our analysis revealed an unexpected relationship between hypodermis and neuron with respect to exopher production. Failure in exopher uptake by the hypodermis might have been expected to result in accumulation of unengulfed exophers, as occurs for apoptotic cell corpses if engulfment fails (*Mangahas and Zhou, 2005*). However, we found that fewer exophers are produced by neurons in *arf-6* and *ced-1* mutants as both exopher and starry night vesicle accumulation is strongly reduced. ARF-6 and CED-1 are required in the hypodermis for this effect as this phenotype is rescued by hypodermis-specific *arf-6* or *ced-1* re-expression, respectively. Moreover, hypodermis-specific RNAi of *arf-6* or its effector *sec-10* also reduce neuronal exopher production, strengthening the conclusion of hypodermal function required for neuronal exophergenesis. Interestingly, depletion of actin, Arp2/3, or loss of *ced-10/Rac* blocks at a later step, accumulating exophers, suggesting that an actin response by the hypodermis is not a key molecular requirement for neuronal exopher production. Further work will be required to understand the precise step in exophergenesis affected by hypodermal recognition. Since early bud frequency appears unaffected in *arf-6* mutants, hypodermal recognition is likely to be required for elaboration of the typical large exopher bud and/or to prevent regression of exopher buds back into the soma prior to bud scission.

CED-1, as a phagocytic receptor, functions in the phagocytic cell, and our CED-1 localization data suggest that hypodermal CED-1 distinguishes between exopher and the attached soma, exclusively associating with the exopher bud. The accumulation of hypodermal CED-1 around the exopher bud but not the neuronal soma, even at relatively early stages of exopher budding, reveals a likely asymmetry in surface membrane components between neuronal soma and exopher. It is interesting to note that mouse cardiomyocytes producing exophers maintain a dedicated pool of attached macrophages to accept produced exophers, suggesting that such close interplay between the exopher ejecting cell and the accepting phagocyte may be a conserved feature of exopher production (*Nicolás-Ávila et al., 2020*). This kind of instructional relationship between phagocyte and neuron is also reminiscent of neuronal pruning by glial cells (*Raiders et al., 2021*). The transfer of expanded polyglutamine aggregates from neurons to associated glia in *Drosophila* further suggests a mechanistic kinship (*Pearce et al., 2015*). Our results with *ttr-52* and *anoh-1* suggest that presentation of phosphatidylserine on the exopher surface contributes to specific exopher recognition, especially given the known role of CED-1 in phosphatidylserine recognition. Further work will be required to fully understand how an exopher-producing cell senses and responds to the associated phagocyte.

## Exopher production may be neuroprotective

The removal of exopher contents from the neuron by phagocytosis is likely to be neuroprotective. Previous work showed that gentle touch response is better preserved in a HttQ128::CFP background when touch neurons had produced an exopher compared to animals in which the touch neurons had not produced an exopher (*Melentijevic et al., 2017*). To extend this analysis, we performed a similar analysis in *arf-6* and *ced-1* mutants in animals expressing mCherry in the touch neurons, comparing gentle touch response to mCherry controls. For both *arf-6* and *ced-1* mutants, we found reduced response to gentle touch in 10-day-old adults (Ad10), but not 5-day-old adults (Ad5), indicating a deficit in old-age neuronal function in mutants in which exopher maturation is compromised (*Figure 8—figure supplement 2*). These results are consistent with a neuroprotective role for exopher production, but caveats remain to this interpretation, as neither ARF-6 nor CED-1 are specific to exophergenesis and may affect neuronal aging in other ways.

## Phagosome maturation is required for the hypodermal processing of neuronal exopher material

A number of trafficking mutants, including *rab-35* and *cnt-1,* allowed full exopher production, but blocked phagosome completion. RAB-35 and CNT-1 have been proposed to be negative regulators of the small GTPase ARF-6, with downregulation of ARF-6 important for the completion of phagocytosis (*Kutscher et al., 2018*). Downregulation of ARF-6 may be required to control relative PI(4,5)P2 and PI(3)P levels, with the proper mix of such lipids required to recruit/activate fission factors such as LST-4/Snx9 and DYN-1/dynamin that complete phagosomal sealing (*Lu et al., 2011*; *Lu et al., 2012*; *Cheng et al., 2015*). Especially high levels of force from outside the neuron may be required to complete exopher phagocytosis if the neuron does not fully sever the soma-exopher connection from

the inside, as might be implied by the long nanotube-like connector frequently observed between the neuronal soma and exopher (*Melentijevic et al., 2017*). A requirement for external forces provided by the phagocyte was recently proposed for severing large lobes produced by *C. elegans* primordial germ cells (*Abdu et al., 2016*).

Consistent with ARF-6 regulation by RAB-35 and CNT-1, *rab-35* and *cnt-1* mutants display the opposite phenotype to the *arf-6* loss-of-function mutant, accumulating rather than lacking exophers, and display an epistasis relationship suggesting that RAB-35 positively regulates CNT-1, and CNT-1 downregulates ARF-6 (*Kutscher et al., 2018*; *Chesneau et al., 2012*; *Shi et al., 2012*; *Allaire et al., 2013*). Indeed, an *arf-6* gain-of-function mutant phenocopies *rab-35* and *cnt-1* mutants. *rab-35* and *cnt-1* mutant exopher accumulation was rescued by hypodermis- but not neuron-specific expression, further indicating that the relevant action of ARF-6 is in the hypodermis, even though it is likely that ARF-6 is expressed in all cells. Subcellular localization results with these proteins were surprising, however; while RAB-35 smoothly labeled the engulfed exopher periphery, CNT-1 and ARF-6 were only visualized as puncta resembling endosomes at the phagosome periphery. Such subcellular distribution could indicate that RAB-35 functions directly on the phagosome, while ARF-6 and CNT-1 could function via endosomal transport or membrane contact rather than assembling directly on the phagosomal membrane.

There are some noteworthy mechanistic differences between exopher phagocytosis and other *C. elegans* phagocytosis events involving RAB-35 and/or CNT-1. First, in the RAB-35/CNT-1-dependent phagocytosis of the *C. elegans* male linker cell, *arf-6* null mutants were epistatic to *rab-35/cnt-1* mutants as we observed, restoring efficient linker cell phagocytosis and degradation (*Kutscher et al., 2018*). However, the *arf-6* single mutant had no effect on linker cell corpse production or disposal, unlike the strong effects *arf-6* mutants have on exopher production. Linker cell phagocytosis was also CED-1 independent, unlike the exopher case (*Kutscher et al., 2018*). The strong effects of ARF-6 and CED-1 single mutants on the exopher system indicate clear mechanistic differences from the linker cell example. One possibility is that ARF-6 and SEC-10/exocyst control CED-1 recycling required for early exopher recognition by the hypodermis. ARF-6 may then also contribute to later steps in phagocytosis as hypodermis-specific depletion of another candidate ARF-6 effector, PPK-1, does not affect exopher production but inhibits the transition from engulfed exopher to exopher-laden phagosome fragments. PPK-1 function is likely required prior phagosomal sealing. We also found that exophers that did form in *arf-6* mutants were more likely than normal to remain soma-attached, similar to our findings after hypodermal *ppk-1* depletion, supporting an additional later role for ARF-6 with PPK-1.

The apparent obligatory progression from exopher-laden phagosome to 'starry night' phagosomal fragmentation prior to degradation of phagosomal content resembles recently described phagosomal resolution processes in the *C. elegans* embryo and in mammalian macrophages (*Ghose and Wehman, 2021*; *Fazeli et al., 2018*; *Levin-Konigsberg et al., 2019*; *Fazeli et al., 2016*). We found that SAND-1/Mon1 was required for exopher-containing phagosome fragmentation. Given the role of SAND-1 as a subunit of the key RAB-7 GDP/GTP exchange factor during phagosome maturation, our results imply an essential role for RAB-7 and its effectors in this process (*Kinchen and Ravichandran, 2010*; *Poteryaev et al., 2010*; *Nordmann et al., 2010*). Recent work in mammalian macrophages indicates a role for Rab7-effector and lipid transfer protein ORPL1 in tethering phagosomal microdomains to the ER in concert with ER localized VAP, acting to reduce PI4P levels and potentially acting as an anchor for the phagosome as pulling forces draw out recycling tubules from the limiting membrane (*Levin-Konigsberg et al., 2019*). It will be of great interest to determine whether such ER tethering is relevant to exopher-laden phagosome resolution and whether failure in this process can explain our results with *sand-1* mutants. Furthermore, we identified a requirement for ARL-8 in exopher-laden phagosome resolution, suggesting a role for kinesin-type motor activity as a driving force in exopher-phagosome tubulation and fragmentation (*Ghose and Wehman, 2021*; *Fazeli et al., 2018*). We also note extensive acquisition of hypodermal LGG-1/LC3 by fragmented, but not whole exopher-laden phagosomes, suggesting either LC3-associated phagocytosis, or fusion of exopher-phagosome fragments with autophagosomes, which may support later content degradation (*Peña-Ramos et al., 2022*). We favor a full autophagosome fusion model since we did not observe LGG-1 association with intact phagosomes as would be expected for LC3-associated phagocytosis. Interestingly, we also did not observe lysosome (LMP-1) marker acquisition on unfragmented phagosomes, which suggests a different order of events from

that described in macrophages, where lysosome fusion to phagosomes precedes fragmentation (*Levin-Konigsberg et al., 2019*).

Taken together, our results identify a set of conserved phagosome resolution mechanisms that operate to break down the large phagosomes containing neuronal exophers, with implications for the ultimate fate of engulfed material, including toxic aggregates. It remains to be determined to what extent neuronally derived toxic materials can escape the hypodermal lysosomal system to reach the cytoplasm, or to be re-secreted to reach additional nearby cells, an important avenue for future analysis. Still, conserved roles for *C. elegans* phagocytosis receptor CED-1, and *Drosophila* counterpart Draper (*Pearce et al., 2015*; *Donnelly et al., 2020*), in the efficient extrusion of disease aggregates underscore the potential importance of related mammalian MEGF10-dependent mechanisms in human pathological aggregate spread (*Singh et al., 2010*; *Fujita et al., 2020*), and invite a new focus on requirements in glial phagocytosis partners in promoting, or clinically addressing, neuropathology.

# Materials and methods

## Plasmids and strains

All *C. elegans* strains were derived originally from the wild-type Bristol strain N2. Worm cultures, genetic crosses, and other *C. elegans* husbandry were performed according to standard methods (*Brenner, 1974*). Mutants used in this study: *anoh-1(tm4762)* (*Li et al., 2015*), *arf-6(tm1447)* (*Shi et al., 2012*), *arl-8(wy271)* (*Klassen et al., 2010*), *ced-1(e1735)* (*Zhou et al., 2001*), *ced-10(n3246)* (*Ellis et al., 1991*), *cnt-1(tm2313)* (*Shi et al., 2012*), *cup-5(ar465)* (*Fares and Greenwald, 2001*), *daf-2(e1370)* (*Kimura et al., 1997*), *rab-35(b1034)* (*Sato et al., 2008*), *ttr-52(tm2078)* (*Wang et al., 2010*), and *sid-1(qt9)* (*Winston et al., 2002*). A complete list of strains used in this study is provided in *Supplementary file 1*.

*C. elegans* expression plasmids utilized the Phyp-7 promoter from the *semo-1* gene for hypodermal expression, or the Pmec-7 promoter from the *mec-7* gene for touch neuron expression (*Philipp et al., 2022*; *Hamelin et al., 1992*; *Li et al., 2016*). Vector details are available upon request. Most cloning was performed using the Gateway in vitro recombination system (Invitrogen, Carlsbad, CA) using in-house-modified versions of hygromycin-resistant and MiniMos-enabled vector pCFJ1662 (gift of Erik Jorgensen, University of Utah, Addgene #51482): pCFJ1662 Phyp7 mNeonGreen GTWY let858 (34B2), pCFJ1662 Phyp7 GTWY mNeonGreen let858 (34H4), pCFJ1662 Phyp7 GTWY oxGFP let858 (35G7), or pCFJ1662 Pmec7 mNeonGreen GTWY let858 (34D4). pDONR221 entry vectors containing coding regions for *lgg-1, rab-5, rab-7, lmp-1, rme-1, aman-2, arf-6, cnt-1, rab-35, ced-10, 2X-FYVE(HRS)*, and *PH-PLCδ* were transferred into hypodermal or neuronal destination vectors by Gateway LR clonase II reaction to generate C-/N- terminal fusions. Single-copy integrations were obtained by MiniMOS technology (*Frøkjaer-Jensen et al., 2008*). The Phyp-7 CED-1ΔC::GFP let-858 plasmid was constructed by amplifying the Phyp-7/*semo-1* promoter from pPD49.26-Phyp-7-sfGFP-Gal3 (gift of Xiaochen Wang, Institute of Biophysics, Chinese Academy of Science) and the *ced-1ΔC* minigene from pZZ645 (gift of Zheng Zhou, Baylor College of Medicine) and inserting into SacII+AgeI digested plasmid pCFJ1662 Pdpy-7 GTWY GFP let858 (30D1), replacing the Pdpy-7 and GTWY segments, using the NEBuilder Gibson assembly kit (New England Biolabs). Phyp7 CED-1(+)::GFP let-858 plasmid (43F6) was constructed by amplifying the *ced-1* minigene from pZZ610 (gift of Zheng Zhou, Baylor College of Medicine) and inserting into BstEII-HF+NgoMIV digested plasmid pCFJ1662 Phyp7 GTWY oxGFP let858 (35G7), replacing the GTWY segment, using the NEBuilder Gibson assembly kit (New England Biolabs).

## Longitudinal exopher tracking

L4 animals expressing touch neuron-specific mCherry (strain ZB5033) were picked onto a seeded 4 cm NGM plate the day before experiment start. ALMRs of free-moving adult D1 or D2 animals were screened at 100–150× on a Kramer FBS10 fluorescent dissecting microscope. Each ALMR exopher-positive animal was picked onto its own individual 35 mm plate and labeled with time of identification, and a sketch of exopher and soma, and descriptions of any exopher vesiculation recorded. Concurrently, exopher-negative animals were transferred to their own 35 mm plates, timestamped and labeled, eventually matching either the total number of exopher-positive animals or 20, whichever was higher. Exopher-negative animals that were not singled out were redistributed to two fresh 4 cm

'communal' plates. All exopher-positive plates were revisited every 30 min to 1 hr. Rechecks were timestamped and any significant changes in exopher appearance or position, particularly onset of SN or exopher fragmentation, were sketched and described. Exopher-negative animals on the communal plates were also rescreened at least once, with each animal being transferred to one of two new 35 mm communal plates. To reduce fluorescence exposure time, initial and subsequent checks were limited to ≤5 s illumination each. D1 and D2 animals that still had intact exophers were rechecked at least three more times. D3: animals were rechecked and singled out as described for D2, except no new exopher-negative 35 mm plates were prepared. Starting D3, all animals would be transferred to a new 35 mm or 4 cm plate every other day. D4, D5: animals were assessed as described for D3. D6–D16: all plates assessed as in D4, except only once per day.

## Exopher and starry night counting

For exopher and starry night frequency measurements, each trial consisted of 50 L4 animals transferred onto a standard OP50 seeded NGM plate. Most experiments were scored on adult day 2, unless otherwise noted. Animals were scored on a Kramer FBS10 fluorescent dissecting microscope after anesthesia via addition of 10 ul of 5 mM levamisole. We scored ALMR neurons for exopher or starry night events in a binary (yes/no) manner. Because exopher and starry night counts are binary, ANOVA and *t*-tests are inappropriate for statistical evaluation. We used the Cochran–Mantel Haenszel (CMH) test to determine p-values.

Exopher identity was based upon size relative to the neuronal soma, with neuron-derived object size larger than ¼ soma (3–5 μm) used for positive identification. Early vesiculation was characterized by the presence of smaller mCherry puncta localized around the exopher, with the exopher still ¼ soma sized or larger. Starry night was characterized by abundant mCherry puncta that appeared near the neuronal soma, often with spread further along the animal. Details about exopher recognition have been published in *Arnold et al., 2020*.

## Confocal microscopy and image analysis

Live animals were mounted on slides using 5% agarose pads and 10 mM levamisole. Multi-wavelength fluorescence images were obtained using a spinning-disk confocal imaging system: Zeiss Axiovert Z1 microscope equipped with X-Light V2 Spinning Disk Confocal Unit (CrestOptics), 7-line LDI Laser Launch (89 North), Prime 95B Scientific CMOS camera (Photometrics), and oil-immersion objectives (×40, ×63, and ×100). Fluorescence images were captured using Metamorph 7.7 software. Z series of optical sections were acquired using a 0.2 μm or 0.5 μm step size.

The 'Integrated Morphometry Analysis' function of Metamorph was used to detect the fluorescent structures that are significantly brighter than the background and to measure total puncta number (referred to as 'structure count') and total fluorescence area (referred to as 'total area') within unit regions. mNeonGreen and mCherry-tagged protein colocalization analysis was performed using the 'Measure colocalization' Application within the Metamorph software. After thresholding, the percentage of green fluorescence area (area A) overlapping with red fluorescence area (area B) in starry night regions was analyzed for each genotype. Colocalization experiments were performed on day 2 adults. The '4D viewer' function of Metamorph was used to detect the overlap of hypo-dermal expressed mNeonGreen markers and neuronal mCherry-labeled intact exophers. After thresholding, we analyzed images for colocalization observing Z-stacks laterally in 3D projection for three-dimensional observation.

## Time-lapse analysis

Time-lapse movies were acquired on a spinning-disk confocal microscope using ×40 and ×63 objectives. For *Videos 1 and 2*, day 2 adults were immobilized by adding 10 ul of 5 mM levamisole to the coverslip before it was placed over the 5% agarose pads. The software was used to mark the stage position for 10 animal's ALMR neuron. Z-stacks were spaced at 0.2 μm. For time-lapse imaging, 30 Z-stacks were acquired every 5–10 min for 2 hr. Data were then inspected using 'Review Multi-Dimensional Data' function to select the optimal plane at each timepoint. The selected sequential planes were then compiled into a AVI file using the 'Make movie' function in Metamorph, with each frame displayed for 1/30th of a second. For *Figure 6—figure supplement 4* and *Videos 3 and 4*, animals were grown at 15°C until the L4 stage, then moved to 20°C for 1 day to reach adult day 1

before scoring. We focused on ALM neurons with no existing exopher bud morphology. Neurons in ZB4857 and RT4231 strains were imaged on average every 4 min for ~4 hr. For this analysis, minimum bud size to be counted was one-tenth the size of source soma.

### Electron microscopy

We prepared D2 hand-picked exopher-positive animals expressing mCherry in the touch neurons for TEM analysis by high-pressure freezing and freeze substitution (HPF/FS) (*Hall et al., 2012*). After HPF in a Baltec HPM-010, we exposed animals to 1% osmium tetroxide, 0.1% uranyl acetate in acetone with 2% water added, held at −90°C for 4 days before slowly warming back to −60°C, −30°C, and 0°C, over a 2-day period. We rinsed the samples several times in cold acetone and embedded the samples into a plastic resin before curing them at high temperatures for 1–2 days. We collected serial thin sections on plastic-coated slot grids and post-stained them with 2% uranyl acetate and then with 1:10 Reynold's lead citrate, and examined with a JEOL JEM-1400 Plus electron microscope with Gatan Orius SC100 bottom-mount digital camera. By observing transverse sections for landmarks such as the second bulb of the pharynx, it was possible to reach the vicinity of the ALM soma before collecting about 1500 serial thin transverse sections, or lengthwise sections for landmarks such as the cuticle before collecting about 1000 serial lengthwise thin sections.

### RNA-mediated interference screening

Hypodermis-specific feeding RNAi was performed in strain ZB4690 by standard methods using HT115 bacteria carrying dsRNA expression plasmid L4440, with or without gene targeting sequences in between the two flanking T7 promoters. We used NGM growth plates supplemented with tetracycline and carbenicillin to select for the L4440 RNAi plasmids, and IPTG (isopropyl β-D-1-thiogalactopyranoside) to induce dsRNA expression. For *act-1*, *act-2*, and *act-3*, scored animals were exposed to RNAi food from L4 to adult day 2. Exposure during earlier developmental stages was lethal. *arx-2* animals were exposed to RNAi food from L1 to adult day 2 before scoring. Scoring the next generation was not possible due to embryonic lethality. For maximum effective knockdown in the ARF-6 effector screen, we scored animals after two full generations of growth on RNAi food. In a typical experiment, 10 adult hermaphrodites were transferred to fresh RNAi plates, allowed to lay eggs for 3 hr, after which the parents were removed. Once the eggs hatched and reached adulthood, 10 adults were again transferred to fresh RNAi plates, allowed to lay eggs, and then removed. Once these eggs hatched and reached L4 stage, 50 animals per trial were transferred again to fresh RNAi plates and scored as day 2 adults. RNAi feeding constructs were obtained from the Ahringer library except for *arf-6, ced-10,* and *unc-16*, in which cDNA sequences were cloned into SacI+HindIII digested L4440 vector after PCR from total N2 cDNA, then re-transformed into HT115 *Escherichia coli* (*Timmons and Fire, 1998*; *Kamath et al., 2003*).

### Touch sensitivity assay

To assay for touch sensitivity, age-synchronized adults were stroked with a single-eyelash hair on alternating anterior and posterior halves of the body (*Zhang et al., 2002*). Reversal was an indication of a positive touch response to anterior stimulation. Animals responding to three of five touches were scored as sensitive, animals responding to two or fewer touches were scored as insensitive. Synchronized animals were tested on adult days 5 and 10. Three biological replicates of 30 animals/replicate were performed.

### Statistical analysis

ALMR exopher occurrence was scored as yes or no (binary), with each trial graphed as a percentage of total. For this type of data, we used Cochran–Mantel Hansel analysis for p-value calculation of three or more biological trials. For hypodermal expressed marker measurements, we used two-tailed *t*-tests. Data were considered statistically different at $p < 0.05$. $*p < 0.05$, $**p < 0.001$, and $***p < 0.0001$.

## Acknowledgements

We thank members of the Grant and Driscoll labs for discussion of data and manuscript feedback. We thank Helen Ushakov for expert microinjection. We thank Zheng Zhou, Erik Jorgensen, Kang Shen, Shai Shaham, Martin Chalfie, and Xiaochen Wang for plasmids and strains. This work was supported

by NIH grant R01AG047101 to MD, BDG, and DHH, NIH Grant R01GM135326 to BDG, NIH grant R37AG056510 to MD, NIH grant R24OD010943 to DHH, NIH grant F31AG066405 to MLA, and NIH grant F31NS101969 to AJS. Some strains were provided by the CGC, which is funded by the NIH Office of Research Infrastructure Programs (P40 OD010440).

## Additional information

### Funding

| Funder | Grant reference number | Author |
|---|---|---|
| National Institutes of Health | R01AG047101 | David H Hall |
| National Institutes of Health | F31AG066405 | Meghan Lee Arnold |
| National Institutes of Health | F31NS101969 | Anna Joelle Smart |
| National Institutes of Health | NIH Office of Research Infrastructure Programs R24OD010943 | David H Hall |
| National Institutes of Health | R01GM135326 | David H Hall |
| National Institutes of Health | R37AG56510 | Monica Driscoll |

The funders had no role in study design, data collection and interpretation, or the decision to submit the work for publication.

### Author contributions

Yu Wang, Conceptualization, Resources, Data curation, Formal analysis, Investigation, Methodology, Writing – review and editing; Meghan Lee Arnold, Anna Joelle Smart, Andres Morera, Ken CQ Nguyen, David H Hall, Conceptualization, Data curation, Formal analysis, Writing – review and editing; Guoqiang Wang, Data curation, Formal analysis, Writing – review and editing; Rebecca J Androwski, Conceptualization, Investigation; Peter J Schweinsberg, Ge Bai, Resources, Data curation, Writing – review and editing; Jason Cooper, Conceptualization, Data curation, Writing – review and editing; Monica Driscoll, Conceptualization, Resources, Funding acquisition, Writing – review and editing; Barth D Grant, Conceptualization, Resources, Data curation, Formal analysis, Supervision, Funding acquisition, Investigation, Methodology, Writing – original draft, Writing – review and editing

### Author ORCIDs

Yu Wang (iD) http://orcid.org/0000-0001-5853-3019
Guoqiang Wang (iD) http://orcid.org/0000-0002-3694-7103
David H Hall (iD) http://orcid.org/0000-0001-8459-9820
Monica Driscoll (iD) http://orcid.org/0000-0002-8751-7429
Barth D Grant (iD) http://orcid.org/0000-0002-5943-8336

### Decision letter and Author response

Decision letter https://doi.org/10.7554/eLife.82227.sa1
Author response https://doi.org/10.7554/eLife.82227.sa2

## Additional files

### Supplementary files
• MDAR checklist
• Supplementary file 1. Table of strains used in this study.

## Data availability

All data generated or analysed during this study are included in the manuscript and supporting files; Source Data files have been provided for Figures 1-8 and their supplements.

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
