## [Editor Report]

This article will be of interest to a wide range of cell biologists working at understanding cell–cell communication. The authors present compelling data showing that large extrusions (exophers) of neuronal cells are taken up by adjacent hypodermal cells and eventually degraded by lysosomes, uncovering an important mechanism for clearing toxic cargo. Mechanistically, the study identifies a number of small GTPases and accessory components, as well as the phagocytic receptor (CED-1).

---

## [Decision Letter]

**Decision letter after peer review:**

Thank you for submitting your article "Large vesicle extrusion from *C. elegans* neurons requires phagocytic interaction via the ARF-6 and CED-1/DRAPER pathways" for consideration by *eLife*. Your article has been reviewed by 3 peer reviewers, and the evaluation has been overseen by a Reviewing Editor and Claude Desplan as the Senior Editor. The following individual involved in review of your submission has agreed to reveal their identity: Marc Freeman (Reviewer #2).

Essential revisions:

1. A common point of concern is whether hypodermal cells are promoting exopher production in neurons. Additional experiments are needed to test this hypothesis, such as site of action rescue for ced-1 and ced-10 (cell-specific rescue), and ruling out that exophers don't simply just retract into the cell body in these mutant backgrounds because they cannot be taken up by hypodermal cells.

2. The authors should also induce exopher production through other means (e.g., expression of mutant Huntingtin protein) and then test any involvement of CED-1. Such experiments would demonstrate broad applicability of the mechanism described herein.

3. The author should experimentally address what is the ultimate fate of a neuron that is unable to produce exophers?

4. An available gain-of-function allele of arf-6 that is likely GTP-locked must be used to confirm the proposed model.

5. The key message of the paper is not clear and modifications of the text and/or title are required. The title suggests that the reader will learn about how ARF-6 and CED-1 control exopher extrusion, but there does not appear to be a substantial amount of data to support this claim.

*Reviewer #1 (Recommendations for the authors):*

The reduced exopher phenotype of arf-6 and ced-1 mutants is very intriguing. The authors show that these proteins function in the hypodermis and propose that they stimulate exopher production by neurons. However, couldn't an alternative hypothesis be that they are required for exopher scission, and that in their absence, exophers are resorbed into the neuronal cell body? This is an important distinction and should be addressed experimentally.

*Reviewer #2 (Recommendations for the authors):*

1. Could the authors perhaps label each graph of the adult age (D2, D3 etc?). I had to read the Figure legends to see that Figure 2F was D2 and Figure 2G was D3. This will help the reader.

2. Can authors provide definitions for I assume other neurons (PVM, ALML, AVM)? I don't see them described in text or figure legend.

3. In Figure 2, the overlap of F-actin and PIP2 appears to be clear. However, the difference in Figure 3 is small (most markers show around 2μm2). More controls would be beneficial. Could the authors use a different control and compare exopher to a non-engulfed exopher instead of exopher to soma? The authors demonstrated that blocking actin assembly would disrupt phagosome formation. This should result in significantly less labeling of phagocytic markers on the periphery of intact ALMR neuron-derived exophers.

4. The authors state in the discussion that the events of exopher breakdown may occur in a different order than macrophages. Because there is no association with lysosomes, it appears that exophers are not phagolysosomes. They do, however, associate with Rab-5 and Rab-7. How acidic are the exophers? Is the exopher labeled with FYVE-GFP? The fact that loss of ARL-8 prevents the formation of smaller vesicles suggests that exophers may be broken down and fused with lysosomes.

5. The authors state that live imaging of these events is challenging. But it would be great to use live imaging, which is the strength of the model system at certain critical points.

6. In the final model Rab5 and Rab7 associate with only the smaller vesicles. Figure 3 implies that they associate also with the exopher.

*Reviewer #3 (Recommendations for the authors):*

1. It seems important for the CED-1/Draper and eat-me signal experiments to perform cell-specific rescue experiments to show that these proteins are acting in the hypodermis and neuron, as suspected. I may have missed it, but I didn't see these studies in the paper. Similarly, for the CED-10 studies, it's important to functionally define the tissue of action.

2. Previously, a gain-of-function allele of arf-6 that is likely GTP-locked was described. Can the authors test the effects of that allele to confirm their model? The diagram in figure 6K would suggest that this would bypass RAB-35 defects. Is that the case?

---

## [Author Response]

Essential revisions:1. A common point of concern is whether hypodermal cells are promoting exopher production in neurons. Additional experiments are needed to test this hypothesis, such as site of action rescue for ced-1 and ced-10 (cell-specific rescue), and ruling out that exophers don't simply just retract into the cell body in these mutant backgrounds because they cannot be taken up by hypodermal cells.

We appreciate the reviewers concerns in requesting further evidence of the function of these components in the hypodermis to influence neuronal exopher production, as we had already shown for *arf-6*, by both hypodermis-specific rescue and hypodermis specific RNAi. We now add data showing rescue of exopher production in *ced-1* mutants by a hypodermis-specific CED-1 transgene (Figure 8B), and have added hypodermis-specific *ced-10* RNAi data (Figure 8F), showing depletion of *ced-10* from the hypodermis is sufficient to induce exopher accumulation. These data further confirm our proposal of a key role for the hypodermis in promoting exopher production.

The other issue brought up by the reviewers was whether the lower exopher rate observed in mutants with a focus in the hypodermis resorbed exopher buds more frequently than controls. We sought to address this issue in time-lapse imaging experiments using *daf-2* mutants, which have an unusually high exopher rate. We compared animals with and without the additional *arf-6* mutation, our best characterized hypodermis-focused exopher suppressor. After analysis of >130 videos lasting several hours each, we did not capture enough exopher events to answer this question directly.

However, we did find a high rate of early bud formation in both genotypes (Figure 8 supp 2; Movies 3 and 4). Our results suggest that neurons begin to form exophers in both genotypes, but the buds rarely progress to completion in *arf-6* mutants, implying a key role for interaction with the hypodermis to elaborate large buds. See text lines 301-307.

2. The authors should also induce exopher production through other means (e.g., expression of mutant Huntingtin protein) and then test any involvement of CED-1. Such experiments would demonstrate broad applicability of the mechanism described herein.

We interpreted this question as an inquiry into whether the neuron-intrinsic exopher inducer was relevant to reliance on hypodermal interaction for exophergenesis, given our use of aggregating mCherry as the inducer. Unfortunately, our Huntingtin expressor lines now display high levels of transgene silencing precluding their use in this experiment. Given this, we switched to a low toxicity GFP-expressing transgene from the Chalfie lab that we found has a detectable level of exopher production from touch neurons, uIs31[Pmec17::GFP]. We found that *arf-6* mutants suppressed exophers in this background as effectively as in previous mCherry experiments (Figure 6E), indicating that our results are not dependent upon the particular transgene marking the touch neurons or the associated expressed protein.

3. The author should experimentally address what is the ultimate fate of a neuron that is unable to produce exophers?

To address this question we measured the function of the mechanosensory touch neurons, using the classic gentle touch response assay in mCherry expressing animals, comparing controls to *arf-6* and *ced-1* mutants. For both mutants, we found reduced response to gentle touch in older adults (Ad10), indicating a deficit in old age neuronal function (Figure 8, supp 2). These results are consistent with exopher production maintaining neuronal health into old age, but interpretation is limited since neither *ced-1* or *arf-6* are specific to exophergenesis and these loci may also affect the animals in other ways. There are currently no known genetic perturbations that are *specific* to the process of exophergenesis, so there is no better current method to address the Reviewer question. We had already published similar results in our 2017 Nature paper that first described exophers, showing that gentle touch response is better preserved in the Htt::Q128::CFP background when touch neurons had produced an exopher as compared to animals in which the Htt::Q128::CFP touch neurons had not produced an exopher.

4. An available gain-of-function allele of arf-6 that is likely GTP-locked must be used to confirm the proposed model.

We obtained a gain-of-function allele of *arf-6* previously described by Shai Shaham, and found that indeed *arf-6* gain-of-function produced more exophers than controls, the opposite of *arf-6* loss-of-function which produces fewer exophers than normal (Figure 6F). These results are fully consistent with our model.

5. The key message of the paper is not clear and modifications of the text and/or title are required. The title suggests that the reader will learn about how ARF-6 and CED-1 control exopher extrusion, but there does not appear to be a substantial amount of data to support this claim.

We have revised the title to “Large vesicle extrusions from *C. elegans* neurons are consumed and stimulated by glial-like phagocytosis activity of the neighboring cell”.

Reviewer #1 (Recommendations for the authors):The reduced exopher phenotype of arf-6 and ced-1 mutants is very intriguing. The authors show that these proteins function in the hypodermis and propose that they stimulate exopher production by neurons. However, couldn't an alternative hypothesis be that they are required for exopher scission, and that in their absence, exophers are resorbed into the neuronal cell body? This is an important distinction and should be addressed experimentally.

We sought to address this issue using *daf-2* mutants, that have an unusually high exopher rate, in time-lapse imaging experiments (Figure S7, Movies 3 and 4). We compared animals with and without the *arf-6* mutation. After analysis of >130 videos lasting several hours each, we did not capture enough exopher events to answer this question directly. However, we did measure a high rate of early bud formation in both genotypes. Our results suggest that neurons begin to form exophers in both genotypes, but the buds rarely progress to completion without ARF-6 function in the hypodermis.

Reviewer #2 (Recommendations for the authors):1. Could the authors perhaps label each graph of the adult age (D2, D3 etc?). I had to read the Figure legends to see that Figure 2F was D2 and Figure 2G was D3. This will help the reader.

We appreciate the reviewer’s concern, but in trying to do this we felt that it cluttered the figures, yet still generally required the reader to read the legend, so we have added adult day 3 labeling on figures where relevant, but did not add Ad2 to all the remaining panels in the paper.

2. Can authors provide definitions for I assume other neurons (PVM, ALML, AVM)? I don't see them described in text or figure legend.

We have added the full definitions of these neuron names to the figure legend.

3. In Figure 2, the overlap of F-actin and PIP2 appears to be clear. However, the difference in Figure 3 is small (most markers show around 2μm2). More controls would be beneficial. Could the authors use a different control and compare exopher to a non-engulfed exopher instead of exopher to soma?

We appreciate the reviewer’s question. We have described in the text that recruitment of maturation markers to intact exophers is small, and appears more adjacent rather than overlapping, suggesting that these markers are acquired by fusion with endosomes and autophagosomes that is much more pronounced during the starry night phase (Figure 4). This is quite different from the strong recruitment of actin at an early phase of phagocytosis. The maturation markers are naturally abundant on endosomes and autophagosomes in the hypodermal cells and some degree of association by chance will occur with the soma regardless of whether the soma has produced an exopher.

Phagosome markers are dynamic and associate at different points of maturation, creating substantial challenge to comparing engulfed vs. non-engufed structures. Our experience is that comparison of the soma from which an exopher is derived to the engulfed exopher is the best control in terms of signal strength and dynamic range.

The authors demonstrated that blocking actin assembly would disrupt phagosome formation. This should result in significantly less labeling of phagocytic markers on the periphery of intact ALMR neuron-derived exophers.

We appreciate the reviewer’s point that such analysis could strengthen our interpretation. We have not directly analyzed the effects of actin disruption on phagocytic marker recruitment because we quantified these markers as only very weakly associated with intact exophers normally, making it quite difficult to measure a reduction. The RNAi-based actin perturbations we used likely only partially reduce actin levels and would likely generate some intermediate level of recruitment effect.

4. The authors state in the discussion that the events of exopher breakdown may occur in a different order than macrophages. Because there is no association with lysosomes, it appears that exophers are not phagolysosomes. They do, however, associate with Rab-5 and Rab-7. How acidic are the exophers?

This is a fascinating question. We have not yet been able to measure exopher pH directly, but we have observed that when GFP and mCherry are both incorporated into exophers, the GFP is visible in intact “early stage” exophers but is often hard to detect during the starry night phase, suggesting that exopher pH starts to drop below pH 6, the pKa of GFP, by the starry night stage. However, even early phagosomes may drop below pH 6. Phagolysosomes should be closer to pH 4. We have tested the improved firePHLY pH sensors developed by Amy Kao, and constructed strains that work well to report pH in other tissues. However, the firePHLY biosensor is based upon fusions to LMP-1/LAMP, which we know does not label exophers or most exopher fragments, precluding its use in the hypodermis to address this issue.

Is the exopher labeled with FYVE-GFP?

We infer that this question is directed at the timing of phagolysosome identity for engulfed exophers. Since a high proportion of the starry night vesicles are positive for 2XFYVE (Figure 4B) (which they should lose by the phagolysosome stage), our data suggest that many starry night vesicles are still immature phagosomes.

The fact that loss of ARL-8 prevents the formation of smaller vesicles suggests that exophers may be broken down and fused with lysosomes.

We agree that there is more work to be done to fully understand at which step ARL-8 functions in exopher-phagosome maturation. While ARL-8 has traditionally been considered as lysosome-enriched, data on the degree to which ARL-8 is enriched on early and late endosomes as opposed to lysosomes are sparse. Two published *C. elegans* papers (PMID: 23485564; 36652947) indicate ARL-8 is also found on RAB-5 positive endosomes and phagosomes, suggesting that ARL-8 could begin action prior to lysosome fusion. We could not find any studies that distinguish between late endosome/phagosome and lysosomes with respect to ARL-8 enrichment.

5. The authors state that live imaging of these events is challenging. But it would be great to use live imaging, which is the strength of the model system at certain critical points.

We agree in principle and would love to have more of this kind of data. Most cases where live imaging is used in *C. elegans* involve imaging events that occur in most or all animals, either continuously or with highly predictable timing, allowing capture of the large number of events required for quantitative interpretation. Since exophergenesis is relatively rare and relatively unpredictable in its timing (10-20% occurrence of a single event sometime during the first 3 days of adulthood), it is much more difficult to capture enough events for interpretation. Adult animals cannot be kept in a healthy state immobilized on slides for more than a few hours at a time. We have found that existing microfluidic devices do not sufficiently immobilize animals for the high-resolution imaging needed for our studies. We have added time lapse studies in the revised manuscript that partly address the dynamics associated with *arf-6* deficits.

6. In the final model Rab5 and Rab7 associate with only the smaller vesicles. Figure 3 implies that they associate also with the exopher.

Model has been updated.

Reviewer #3 (Recommendations for the authors):1. It seems important for the CED-1/Draper and eat-me signal experiments to perform cell-specific rescue experiments to show that these proteins are acting in the hypodermis and neuron, as suspected. I may have missed it, but I didn't see these studies in the paper. Similarly, for the CED-10 studies, it's important to functionally define the tissue of action.

We appreciate the reviewers concerns in providing further evidence of the function of these components in the hypodermis to influence neuronal exopher production, as we had already shown for *arf-6*, by both hypodermis-specific rescue and hypodermis-specific RNAi. We now add data showing rescue of exopher production in *ced-1* mutants by a hypodermis-specific CED-1 transgene, and have added hypodermis-specific *ced-10* RNAi data, showing depletion of *ced-10* from the hypodermis is sufficient to induce exopher accumulation. These data further confirm our proposal of a key role for the hypodermis in promoting exopher production.

2. Previously, a gain-of-function allele of arf-6 that is likely GTP-locked was described. Can the authors test the effects of that allele to confirm their model? The diagram in figure 6K would suggest that this would bypass RAB-35 defects. Is that the case?

We obtained a gain-of-function allele of *arf-6* previously described by Shai Shaham, and found that indeed *arf-6* gain-of-function produced more exophers than controls, the opposite of *arf-6* loss-of-function which produces fewer exophers than normal (Figure 6F). These results are fully consistent with our model.

There was some confusion as to the relationship between RAB-35, CNT-1, and ARF-6, likely due to our text in the results and the way we depicted the relationship in the figure diagram. We have revised the text and figure to help clarify. We note that since RAB-35 and CNT-1 are negative regulators of ARF-6, GTP-locked ARF-6 would phenocopy *rab-35* and *cnt-1* mutants, rather than bypass them. It does (Figure 6F). Another prediction is that *arf-6(0)* would suppress the phenotypes of *rab-35* and *cnt-1*, which was shown in original Figure 6E (now Figure 6G).

We revised the text to clarify and better make these points.